Three new species of the spider genus Naphrys Edwards (Araneae, Salticidae) under morphology and molecular data with notes in the distribution of Naphrys acerba (Peckham & Peckham) from Mexico

Maldonado-Carrizales Juan 1
Valdez-Mondragón Alejandro 2
Jiménez-Jiménez María L. 2
Ponce-Saavedra Javier javier.ponce@umich.mx 1
1 Faculty of Biology, Universidad Michoacana de San Nicolás de Hidalgo , Morelia , Michoacan , Mexico
2 Arachnological Collection, Centro de Investigaciones Biológicas del Noroeste , La Paz , Baja California Sur , Mexico
Shrestha Jiban
Electronic publication date: 2025 Jan 31
Publication date: 2025
Volume: 13
Electronic Location ID: e18775
Received 2024 Aug 5; Accepted 2024 Dec 9
Copyright: ©2025 Maldonado-Carrizales et al.
Copyright year: 2025
Copyright holder: Maldonado-Carrizales et al.
License: This is an open access article distributed under the terms of the Creative Commons Attribution License, which permits unrestricted use, distribution, reproduction and adaptation in any medium and for any purpose provided that it is properly attributed. For attribution, the original author(s), title, publication source (PeerJ) and either DOI or URL of the article must be cited.
License URL: https://creativecommons.org/licenses/by/4.0/

Keywords: Integrative taxonomy, Neotropical, Euophryini, DNA barcode, Jumping spiders, Biodiversity, Biogeography

Funding: The Consejo Nacional de Humanidades, Ciencia y Tecnología (CONHACYT) of Mexico through the scholarship no. 790303 The work is funded by the Consejo Nacional de Humanidades, Ciencia y Tecnología (CONHACYT) of Mexico through the scholarship no. 790303. The funders had no role in study design, data collection and analysis, decision to publish, or preparation of the manuscript.

==============================
Herein, we describe three new species of the spider genus Naphrys Edwards, 2003 from Mexico: Naphrys echeri sp. nov., Naphrys tecoxquin sp. nov., and Naphrys tuuca sp. nov. An integrative taxonomic approach was applied, utilizing data from morphology, ultra-morphology, the mitochondrial gene COI, and distribution records. Four molecular methods for species delimitation were implemented under the corrected p-distance Neighbor-Joining (NJ) criteria: (1) Assemble Species by Automatic Partitioning (ASAP); (2) general mixed Yule coalescent (GMYC); (3) Bayesian Poisson tree process (bPTP); and (4) multi-rate Poisson tree process (mPTP). Both morphological and molecular data supported the delimitation and recognition of the three new species. The average interspecific genetic distance (p-distance) within the genus Naphrys is 14%, while the intraspecific genetic distances (p-distance) is <2% for most species. We demonstrate that the natural distribution of Naphrys is not restricted to the Nearctic region. Furthermore, the reported localities herein represent the first with precise locations in the country for Naphrys acerba. In addition, a taxonomic identification key is provided for the species in the genus.

Introduction

The spider family Salticidae, comprised more than 6,700 described species, represents the most diverse spider family worldwide (WSC, 2024). One of the largest groups within this family is the tribe Euophryini, containing over 1,000 species within 116 genera (Edwards, 2003; Maddison, 2015; Zhang & Maddison, 2015). Euophryine species have a global distribution, primarily found in tropical regions (Zhang & Maddison, 2015; Maddison, 2015). They exhibit a remarkable uniformity in body shape, with elongate or ant-like forms uncommon, their genitalia also share some particular characteristics: the male palp typically has a simple spiral embolus, and the epigynum has windows framed by circular folds, presumably guiding the embolus during mating (Maddison, 2015).

The taxonomy of the tribe is encumbered by common morphological convergences and reversals, despite attempts at species delimitation using both morphological and molecular data. This taxonomic confusion is further compounded by the relative simplicity of Euophryini genitalia, which exhibit limited interspecific variation and hinder even genus-level identification (Zhang & Maddison, 2015).

According to Edwards (2003), most Euophryine species in the Nearctic region are small (less than 5 mm long) with compact bodies. These species often exhibit cryptic coloration (browns or grays) and possess a moderate number of setae on their bodies. The genus Naphrys Edwards, 2003 is as a clear representative of this group. Naphrys currently includes four described species restricted to North America: Naphrys acerba (Peckham & Peckham, 1909), Naphrys bufoides (Chamberlin & Ivie, 1944), Naphrys pulex (Hentz, 1846), and Naphrys xerophila (Richman, 1981) are all found in the United States. Additionally, N. pulex extends into Canada, and N. acerba has been reported in Mexico (Richman, 1981; Edwards, 2003; WSC, 2024).

In Mexico, the distribution of N. acerba is reported in the northeastern region, but precise locations remain unclear (Richman, 1981). Nevertheless, diverse sources (Ibarra-Núñez, Maya-Morales & Chamé-Vázquez, 2011; Maddison, 2015; Maldonado-Carrizales & Ponce-Saavedra, 2017) mention the presence of the genus in different parts of Mexico without assigning known species. This highlights the limited taxonomic knowledge about this genus in the country.

Modern taxonomy enlists a wide variety of methods and different lines of evidence to analyze and delimit lineages, as morphological evidence alone can be extremely difficult or impossible to delimit species in some cases (Hebert et al., 2003; Carstens et al., 2013; Luo et al., 2018; Nolasco & Valdez-Mondragón, 2022; Hedin & Milne, 2023). This approach recognizes the limitations of relying solely on morphology.

DNA analysis has become a crucial tool in species delimitation due to its objectivity. Unlike morphology that can be subjective and influenced by the environment, DNA offers a standardized and quantifiable measure of evolutionary divergence (Fujita et al., 2012). Nevertheless, delineating or delimiting spider species based only on molecular data is insufficient and incorrect (Hamilton, Formanowicz & Bond, 2011).

The combined use of morphological and molecular data is becoming increasingly important for species delimitation in spiders. This approach is particularly valuable in families like Salticidae, where similar appearances and sexual characteristics make traditional classification methods challenging (Trębicki et al., 2021; Cala-Riquelme, Bustamante & Salgado, 2022; Maddison et al., 2022; Courtial et al., 2023; Kumar, Gupta & Sharma, 2024; Lin, Yang & Zhang, 2024; Phung et al., 2024). Similar successes have been achieved in other spider groups such as Mygalomorphae (Hamilton et al., 2014; Ortiz & Francke, 2016; Candia-Ramírez & Francke, 2021; Ferretti, Nicoletta & Soresi, 2024), Synspermiata (Valdez-Mondragón et al., 2019; Navarro-Rodríguez & Valdez-Mondragón, 2020; Navarro-Rodríguez & Valdez-Mondragón, 2024) and Araneoidea (Hedin & Milne, 2023). The combined use of methods has resulted in robust characterizations of species boundaries.

The integrative taxonomy approach has emerged to address shortcomings of each method individually, using multiple data sources and disciplines in a complementary way to identify and delimit species or lineages. In other words, integrative taxonomy is the method that aims to delimit species, the fundamental units of biodiversity, from different and complementary perspectives (Dayrat, 2005; DeSalle, Egan & Siddall, 2005; Padial et al., 2010; Padial & De La Riva, 2010).

While integrative taxonomy has been applied in various biological groups, its use in spider research remains limited (Bond et al., 2021). This highlights the potential for further exploration of integrative taxonomy within spider systematics.

In this study, we employ integrative taxonomy to describe three new species of the genus Naphrys. This approach utilizes morphological characters, ultra-morphology, and molecular data analyzed using both genetic-distance and tree-based methods for species delimitation. As there is no single species concept, in this work we employ the unified species concept, which is a flexible framework that incorporates elements from various species concepts such as the biological, ecological, evolutionary, and phylogenetic concepts, to delimit species based on their status as separately evolving metapopulation lineages (De Queiroz, 2007; Schlick-Steiner et al., 2010). We also consider the biogeographical distribution records of the new species. Finally, we provide a taxonomic identification key for the species of the genus and accurate distribution data for N. acerba in northeastern Mexico.

Materials & Methods

The specimens were collected and preserved in both 96% ethanol for molecular analyses and 80% ethanol with complete field data labels for morphological studies. Type specimens are deposited at two biological collections: Colección de Arácnidos e Insectos, Centro de Investigaciones Biológicas del Noroeste, S.C. (CARCIB), La Paz, Baja California Sur, Mexico, and Colección Aracnológica de la Facultad de Biología de la Universidad Michoacana (CAFBUM), Morelia, Michoacán, Mexico. The specimens were collected under the document SPARN/DGVS/074492/24, Scientific Collector Permit from the Secretaría de Medio Ambiente y Recursos Naturales (SEMARNAT), Mexico, provided to Margarita Vargas Sandoval (Director and Head curator of the CAFBUM, Faculty of Biology, Entomology Laboratory, Universidad Michoacana de San Nicolás de Hidalgo). For morphological descriptions, specimens were observed using an Amscope SM1TZ-RL-10MA stereomicroscope. All measurements are in millimeters (mm). Epigyna were dissected, manually cleaned, and temporarily cleared with clove oil following the method described by Levi (1965), after digesting the internal epigynal soft tissues with KOH 10%. Left male palps were dissected and cleaned manually using hypodermic needles and a small brush. Both genitalia were observed under a transmitted light microscope Axiostar Plus Carl Zeiss. Habitus and genitalia photographs were obtained using separate setups, an Amscope MU1000 camera attached to an Amscope SM1TZ-RL-10MA stereomicroscope for habitus images, and a transmitted light microscope (Axiostar Plus Carl Zeiss) for genitalia. Photographs were processed with the Helicon focus v8.2.2 program and edited using Adobe Photoshop CS6. The distribution map was created using QGIS v3.32 ‘Lima’. Biogeographic province data (.shp) were obtained from the proposed boundaries by Morrone, Escalante & Rodríguez-Tapia (2017), and Escalante, Rodríguez-Tapia & Morrone (2021). Boundary data (.shp files) were sourced from USGS (2021). Finally, the topographic base layer used was ‘ESRI Topo’ via the subprogram XYZ Tiles in QGIS. For scanning electron microscopy (SEM), morphological structures were dissected, cleaned manually, dehydrated in absolute ethanol, critical-point dried with samdri-PVT-3B equipment, and then covered with gold:palladium in a 60:40 proportion. The structures were examined under low vacuum in a Hitachi S-3000N SEM. Measurements on electron micrographs are in micrometers (µm). Morphological nomenclature mostly follows Ramírez (2014) and Zhang & Maddison (2015), with abbreviations used in the description and figures as follows: AER, anterior eyes row; PER, posterior eyes row; ALE, anterior lateral eye; AME, anterior median eye; PLE, posterior lateral eye; PME, posterior median eye; OQ, ocular quadrangle; S, spermatheca; CD, copulatory duct; W, window of epigynum; CO, copulatory openings; FD, fertilization duct; MS, median septum; RTA, retrolateral tibial apophysis; E, embolus; ED, embolic disc; SP, sperm pore; T, tegulum; TL, tegular lobe; RSDL, retrolateral sperm duct loop; VTA, ventral tibial apophysis; PED, process on embolic disc.

Taxon sampling

The molecular analyses were carried out with a total of 110 specimens, including one undescribed species of Naphrys and three new Naphrys species described herein. Because this study it is not a phylogenetic analysis, we use only one outgroup taxon to root the trees, Corticattus latus Zhang & Maddison, 2012 which represents the genus most closely related to Naphrys according with Zhang & Maddison (2015) (Table 1).

Table 1 Specimens used in the molecular analyses under COI, DNA voucher numbers, localities, and GenBank/BOLD accession numbers.

Specie	DNA voucher numbers	Locality	GenBank/BOLD accesion number	Source	
Naphrys pulex	Npulex_CAN1	Canada: Ontario	HM880192	Blagoev et al. (2016)	
	Npulex_CAN2	Canada: Wellintong	GU682819	Blagoev et al. (2016)	
	Npulex_CAN3	Canada: Wellintong	GU682817	Blagoev et al. (2016)	
	Npulex_CAN4	Canada: Wellintong	GU682816	Blagoev et al. (2016)	
	Npulex_CAN5	Canada: Wellintong	GU682814	Blagoev et al. (2016)	
	Npulex_CAN6	Canada: Wellintong	GU682836	Blagoev et al. (2016)	
	Npulex_CAN7	Canada: Wellintong	ARONT843-18	Blagoev et al. (2016)	
	Npulex_CAN8	Canada: Wellintong	ARONT876	Ratnasingham & Hebert (2013)	
	Npulex_CAN9	Canada: Ontario	ARONT917	Ratnasingham & Hebert (2013)	
	Npulex_CAN10	Canada: Wellintong	ARONT947	Ratnasingham & Hebert (2013)	
	Npulex_CAN11	Canada: Ontario	KP646979	Blagoev et al. (2016)	
	Npulex_CAN12	Canada: Ontario	KP656563	Blagoev et al. (2016)	
	Npulex_CAN13	Canada: Ontario	MG049224	Ratnasingham & Hebert (2013)	
	Npulex_CAN14	Canada: Ontario	ARONZ306	Ratnasingham & Hebert (2013)	
	Npulex_CAN15	Canada: Ontario	ARONZ331	Ratnasingham & Hebert (2013)	
	Npulex_CAN16	Canada: Ontario	ARONZ571	Ratnasingham & Hebert (2013)	
	Npulex_CAN17	Canada: Ontario	HQ924681	Blagoev et al. (2016)	
	Npulex_CAN18	Canada: Ontario	HQ924683	Blagoev et al. (2016)	
	Npulex_CAN19	Canada: Nova Scotia	GU683271	Blagoev et al. (2016)	
	Npulex_CAN20	Canada: Nova Scotia	GU683271	Blagoev et al. (2016)	
	Npulex_CAN21	Canada: Ontario	MF816087	deWaard et al. (2019)	
	Npulex_CAN22	Canada: Nova Scotia	KP652066	Blagoev et al. (2016)	
	Npulex_CAN23	Canada: Quebec	KP646121	Blagoev et al. (2016)	
	Npulex_CAN24	Canada: Ontario	MF808927	deWaard et al. (2019)	
	Npulex_CAN25	Canada: Ontario	MF816952	deWaard et al. (2019)	
	Npulex_CAN26	Canada: Ontario	KP651428	Blagoev et al. (2016)	
	Npulex_CAN27	Canada: Ontario	KP648109	Blagoev et al. (2016)	
	Npulex_CAN28	Canada: Ontario	MF810509	deWaard et al. (2019)	
	Npulex_CAN29	Canada: Ontario	ELPCG2846	Ratnasingham & Hebert (2013)	
	Npulex_CAN30	Canada: Ontario	ELPCG2847	Ratnasingham & Hebert (2013)	
	Npulex_CAN31	Canada: Ontario	ELPCG3050	Ratnasingham & Hebert (2013)	
	Npulex_CAN32	Canada: Ontario	ELPCG3523	Ratnasingham & Hebert (2013)	
	Npulex_CAN33	Canada: Ontario	ELPCG3524	Ratnasingham & Hebert (2013)	
	Npulex_CAN34	Canada: Ontario	ELPCG3525	Ratnasingham & Hebert (2013)	
	Npulex_CAN35	Canada: Ontario	ELPCG3599	Ratnasingham & Hebert (2013)	
	Npulex_CAN36	Canada: Ontario	ELPCG5003	Ratnasingham & Hebert (2013)	
	Npulex_CAN37	Canada: Ontario	ELPCG5472	Ratnasingham & Hebert (2013)	
	Npulex_CAN38	Canada: Ontario	ELPCG6449	Ratnasingham & Hebert (2013)	
	Npulex_CAN39	Canada: Ontario	ELPCG7399	Ratnasingham & Hebert (2013)	
	Npulex_CAN40	Canada: Ontario	ELPCG7401	Ratnasingham & Hebert (2013)	
	Npulex_CAN41	Canada: Ontario	ELPCG8416	Ratnasingham & Hebert (2013)	
	Npulex_CAN42	Canada: Ontario	ELPCG8449	Ratnasingham & Hebert (2013)	
	Npulex_CAN43	Canada: Ontario	ELPCG8644	Ratnasingham & Hebert (2013)	
	Npulex_CAN44	Canada: Ontario	ELPCH2306	Ratnasingham & Hebert (2013)	
	Npulex_CAN45	Canada: Ontario	MG048013	deWaard et al. (2019)	
	Npulex_CAN47	Canada: Nova Scotia	KP649884	Blagoev et al. (2016)	
	Npulex_CAN48	Canada: Nova Scotia	KP654153	Blagoev et al. (2016)	
	Npulex_CAN49	Canada: Ontario	KP652349	Blagoev et al. (2016)	
	Npulex_CAN50	Canada: Nova Scotia	MF809281	deWaard et al. (2019)	
	Npulex_CAN51	Canada: Nova Scotia	MF813033	deWaard et al. (2019)	
	Npulex_CAN52	Canada: Ontario	OPPKG2671	Ratnasingham & Hebert (2013)	
	Npulex_CAN53	Canada: Ontario	OPPOG1872	Ratnasingham & Hebert (2013)	
	Npulex_CAN54	Canada: Ontario	OPPZE1286	Ratnasingham & Hebert (2013)	
	Npulex_CAN55	Canada: Wellintong	KP647608	Blagoev et al. (2016)	
	Npulex_CAN56	Canada: Ontario	KM839902	Blagoev et al. (2016)	
	Npulex_CAN57	Canada: Ontario	JN308610	Blagoev et al. (2016)	
	Npulex_CAN58	Canada: Ontario	JN308622	Blagoev et al. (2016)	
	Npulex_CAN59	Canada: Ontario	JN308631	Blagoev et al. (2016)	
	Npulex_CAN60	Canada: Ontario	JN308807	Blagoev et al. (2016)	
	Npulex_CAN61	Canada: Ontario	JN308822	Blagoev et al. (2016)	
	Npulex_CAN62	Canada: Ontario	RARBB197	Ratnasingham & Hebert (2013)	
	Npulex_CAN63	Canada: Ontario	RARBB202	Ratnasingham & Hebert (2013)	
	Npulex_CAN64	Canada: Ontario	DQ127443	Barrett & Hebert (2005)	
	Npulex_CAN65	Canada: Ontario	DQ127431	Barrett & Hebert (2005)	
	Npulex_CAN66	Canada: Ontario	RBGBB303	Ratnasingham & Hebert (2013)	
	Npulex_CAN67	Canada: Ontario	ROUGE2474	Ratnasingham & Hebert (2013)	
	Npulex_CAN68	Canada: Ontario	KT707577	Telfer et al. (2015)	
	Npulex_CAN69	Canada: Ontario	KT707910	Telfer et al. (2015)	
	Npulex_CAN70	Canada: Ontario	KT706489	Telfer et al. (2015)	
	Npulex_CAN71	Canada: Ontario	KT619474	Telfer et al. (2015)	
	Npulex_CAN72	Canada: Ontario	MG048049	Ratnasingham & Hebert (2013)	
	Npulex_CAN73	Canada: Ontario	MG046695	Ratnasingham & Hebert (2013)	
	Npulex_CAN74	Canada: Ontario	MG044990	Ratnasingham & Hebert (2013)	
	Npulex_CAN75	Canada: Ontario	HQ977049	Blagoev et al. (2016)	
	Npulex_CAN76	Canada: Ontario	KP650393	Blagoev et al. (2016)	
	Npulex_CAN77	Canada: Ontario	KP656197	Blagoev et al. (2016)	
	Npulex_CAN78	Canada: Ontario	KP646924	Blagoev et al. (2016)	
	Npulex_CAN79	Canada: Ontario	KP656484	Blagoev et al. (2016)	
	Npulex_CAN80	Canada: Ontario	KP649929	Blagoev et al. (2016)	
	Npulex_CAN81	Canada: Ontario	MG046512	Ratnasingham & Hebert (2013)	
	Npulex_CAN82	Canada: Ontario	MG043132	Ratnasingham & Hebert (2013)	
	Npulex_CAN83	Canada: Ontario	MF815739	deWaard et al. (2019)	
	Npulex_CAN84	Canada: Ontario	MG509225	deWaard et al. (2019)	
	Npulex_CAN85	Canada: Ontario	MG509777	deWaard et al. (2019)	
	Npulex_CAN86	Canada: Ontario	KP656878	Blagoev et al. (2016)	
	Npulex_CAN87	Canada: Ontario	KP656232	Blagoev et al. (2016)	
	Npulex_USA2	United States: Texas	BBUSE1504 (BIOUG01877-H01)	Ratnasingham & Hebert (2013)	
	Npulex_USA3	United States: Tennessee	GMGSQ008 (BIOUG03453-H09)	Ratnasingham & Hebert (2013)	
	Npulex_USA4	United States: Tennessee	GMGST563 (BIOUG04938-E01)	Ratnasingham & Hebert (2013)	
	Npulex_USA5	United States: Washington	GMNCF099	Ratnasingham & Hebert (2013)	
	Npulex_USA6	United States: Unknown	OR235169	NCBI (2024)	
Naphrys xerophila	Nxerophila_USA1	United States: Texas	BBUSE1415 (BIOUG01637-H07)	Ratnasingham & Hebert (2013)	
Naphrys sp.	Nsp_USA10	United States: High Appalachian Mountains	OR174102	Caterino & Recuero (2023)	
	Nsp_USA11	United States: High Appalachian Mountains	OR173350	Caterino & Recuero (2023)	
	Nsp_USA8	United States: High Appalachian Mountains	OR174487	Caterino & Recuero (2023)	
	Nsp_USA9	United States: High Appalachian Mountains	OR174414	Caterino & Recuero (2023)	
Naphrys echeri sp. nov.	Necheri_MEX11	Mexico: Michoacán	PP123908	Present study	
	Necheri_MEX58	Mexico: Michoacán	PP123905	Present study	
	Necheri_MEX71	Mexico: Michoacán	PP123902	Present study	
	Necheri_MEX73	Mexico: Michoacán	PP123909	Present study	
	Necheri_MEX74	Mexico: Michoacán	PP123903	Present study	
	Necheri_MEX8	Mexico: Michoacán	PP123900	Present study	
	Necheri_MEX9	Mexico: Michoacán	PP123901	Present study	
Naphrys tecoxquin sp. nov.	Ntecoxquin_MEX109	Mexico: Jalisco	PP123899	Present study	
	Ntecoxquin_MEX54	Mexico: Jalisco	PP123898	Present study	
	Ntecoxquin_ MEX 56	Mexico: Jalisco	PP123906	Present study	
Naphrys tuuca sp. nov.	Ntuuca_ MEX 52	Mexico: Nayarit	PP123904	Present study	
	Ntuuca_ MEX 76	Mexico: Nayarit	PP123910	Present study	
	Ntuuca_ MEX 98	Mexico: Nayarit	PP123907	Present study	
Corticattus latus	Clatus_DomRep	Dominican Republic: Pedernales	KC615698	Zhang & Maddison (2015)	

DNA extraction, amplification, and sequencing

The DNA was isolated separately from all eight legs of 13 individual specimens, using proteinase K/phenol/chloroform following the protocol by Hillis et al. (1996). Briefly, all eight legs of a single spider were incubated at 60 °C for 24 h with a digestion buffer containing 400 µL saline solution, 45 µL of 1% SDS solution, and 5 uL of proteinase K. After digestion, 200 µL of Phenol and 200 µL of isoamyl chloroform was added and shaken vigorously. Afterwards, samples were centrifuged at 12,000 rpm for 10 min. Once finished, 400 µL of upper aqueous phase was recovered, repeating the phenol/chloroform washes once more. Once the phenol/chloroform washes were done, 200 µL of phenol was added to the mixture, shaken gently, and then centrifuged immediately at 12,000 rpm for 10 min. 300 µL of upper aqueous phase was recovered and 750 µL of cold (−20 °C) absolute ethanol was added. The mixture was then shaken gently and incubated for 12 h at −20 °C. Once incubated, it was centrifuged at 13,000 rpm for 20 min, and the ethanol was decanted by inversion, avoiding losing the bottom pellet. 600 µL of cold 70% ethanol (−20 °C) was then added and centrifuged at 13,000 rpm for 20 min, with ethanol decanting by inversion while avoiding losing the bottom pellet. Finally, drying in a vacuum centrifuge was performed at 60 °C for 10 min. Once the vial is dry, DNA is suspended in 50 µL of distilled water and stored at −20 °C. After DNA extraction, the mitochondrial gene Cytochrome Oxidase subunit 1 (COI), proposed by Folmer et al. (1994), was amplified (LCO1498 and HCO2198). Amplifications were carried out in a GeneAmp PCR System 2700 thermal cycler, in a total volume of 25.9 µL: 1.66 µL Buffer (5X), 1.5 µL MgCl2 (50 mM), 1.25 µL LCOI1498 (10 µM), 1.25 µL HCOI2198 (10 µM), 0.23 µL Taq (5U/µL), 0.875 µL dNTP’s (10 mM), 1 µL BSA (1.25 mg/µL), 16.135 µL H2O, 2 µL DNA. The PCR was set up as follows: an initial step for 1 min 30 s at 95 °C; 35 amplification cycles of 30 s at 94 °C (denaturation), 30 s at 50 °C (annealing), 45 s at 72 °C (elongation), and final elongation of 10 min at 72 °C. PCR products were checked via gel electrophoresis to analyze length and purity on 1% agarose gels with a molecular marker of 1,000 bp.

DNA extractions were carried out at the Laboratorio de Biología Acuática “J. Javier Alvarado Díaz”, while PCR amplifications were carried out at the Centro Multidiciplinario de Estudios en Biotecnología (CMEB), both at the Universidad Michoacana de San Nicolás de Hidalgo (UMSNH) in Morelia, Michoacán, Mexico. Purification and sanger sequencing in both directions were carried out in Psomagen, Maryland, United States.

Sequence editing and alignment

The sequences were visualized in Geneious Prime v.2023.2.1 (Geneious Prime, 2023) and then manually edited using the BioEdit v. 7.7.1 program (Hall, 1999). After saving in FASTA format (.fas), the sequences were aligned using MAFFT v. 7 (Katoh & Toh, 2008) with default parameters on the MAFFT online server (https://mafft.cbrc.jp/alignment/server/).

Molecular analysis and species delimitation

Four different molecular delimitation methods were employed using the corrected p-distances neighbor-joining (NJ) as initial criteria: (1) ASAP (Assemble Species by Automatic Partitioning) (Puillandre, Brouillet & Achaz, 2021), (2) GMYC (General Mixed Yule Coalescent) (Pons et al., 2006), (3) bPTP (Bayesian Poisson Tree Process) (Zhang et al., 2013), and (4) mPTP (multi-rate Poisson tree processes) (Kapli et al., 2017).

p-distances neighbor-joining (NJ) criteria

MEGA v.10.0.5 (Kumar et al., 2018) was used to construct the genetic distance tree, using the following parameters: number of replicates = 1,000, bootstrap support values = 1,000 (significant values ≥ 50%), substitution type = nucleotide, model = p-distance, substitutions to include = d: transitions + transversions, rates among sites = gamma distributed with invariant sites (G+I), missing data treatment = pairwise deletion.

Assemble species by automatic partitioning (ASAP)

This method is an ascending hierarchical clustering algorithm that analyzes single-locus DNA barcode datasets. It iteratively merges sequences with the highest pairwise similarity into progressively larger clusters. Additionally, ASAP retains information on all potential clustering steps, resulting in a comprehensive series of partitions representing putative species groupings within the data. Subsequently, ASAP calculates a probability score for each partition based on the within-group sequence similarity compared to between-group similarity. Finally, the method identifies the partitions with the highest probability scores as the most likely species-level groupings and constructs a species partition tree reflecting the hierarchical relationships among these putative species (Puillandre, Brouillet & Achaz, 2021). ASAP analyses were run on the online platform (https://bioinfo.mnhn.fr/abi/public/asap/) using Kimura (K80) distance matrices and configured under following parameters: substitution model = p-distances, probability = 0.01, best scores = 10, fixed seed value = −1.

General mixed yule coalescent (GMYC)

The GMYC method (Fujisawa & Barraclough, 2013) is a statistical framework employed for species delimitation using single-locus DNA barcode data. This approach utilizes single time thresholds to define species boundaries within a Maximum Likelihood context, relying on ultrametric trees as input (Ortiz & Francke, 2016; Nolasco & Valdez-Mondragón, 2022). Ultrametric trees were generated in this study through phylogenetic analyses performed in BEAUti and BEAST v.2.7.6 software (Bouckaert et al., 2019). A Yule Process tree prior was implemented during the analysis to account for lineage diversification patterns. Furthermore, an optimized relaxed molecular clock model was applied, incorporating the estimated evolutionary model for the COI gene (GTR + I + G). To ensure robustness of the phylogenetic inference, five independent BEAST analyses were executed, each running for 80 million iterations. Convergence of these analyses was subsequently evaluated using Tracer v1.6 (Rambaut & Drummond, 2003–2013), with a minimum threshold of 200 for the Effective Sample Sizes (ESS). Following this, Tree Annotator 2.6.0 (part of the BEAST package) was employed to generate maximum likelihood trees representing the most likely evolutionary histories. The first 25% of each independent run was discarded as burn-in to account for potential initial biases in the MCMC chains. Finally, the GMYC method was implemented through the online platform (https://species.h-its.org/gmyc/) (Fujisawa & Barraclough, 2013).

Bayesian Poisson tree processes (bPTP)

bTPT operates within a Bayesian framework, accounting for uncertainties in both the phylogenetic tree’s branch lengths and potential species assignments. This method assumes a Poisson process for speciation events along the tree branches and incorporates branch lengths reflecting sequence divergences. Considering this information and its inherent uncertainties, bPTP estimates posterior probabilities for various candidate species partitions within the data, which represent the likelihood of each partition accurately reflecting true species boundaries (Zhang et al., 2013). In this work, Bayesian and maximum likelihood variants were carried out on the online platform (https://species.h-its.org/ptp/), using following options: rooted tree, MCMC = 1000000, thinning = 100, burn-in = 0.1, seed = 123. The resulting trees were edited in FigTree 1.4.4 (Rambaut, 2018). Congruence integration criteria were employed to delimit different species. This approach compares evidence across multiple methods, resulting in more robust species delimitations and better supported species hypotheses (e.g., DeSalle, Egan & Siddall, 2005; Pons et al., 2006; Navarro-Rodríguez & Valdez-Mondragón, 2020; Valdez-Mondragón, 2020; Nolasco & Valdez-Mondragón, 2022).

Multi-rate Poisson tree processes

Multi-rate poisson tree processes (mPTP) uses a non-homogeneous Poisson process model. This approach allows for the estimation of distinct rate multipliers for individual branches within the phylogenetic tree, recognizing potential heterogeneity in evolutionary rates across lineages. ML tree estimation was used to identify branch-specific rate multipliers, and Markov chain Monte Carlo (MCMC) simulations were employed to integrate over the uncertainty associated with these estimates (Kapli et al., 2017). By identifying statistically significant shifts in diversification rates along the tree generated from our ML analysis, mPTP pinpoints potential species boundaries, specifically taking into account lineages that have undergone evolution at disparate paces. This analysis was carried out on the online platform (http://mptp.h-its.org/).

Zoobank

The electronic version of this article in Portable Document Format (PDF) will represent a published work according to the International Commission on Zoological Nomenclature (ICZN). Hence, the new names contained in the electronic version are effectively published under that Code from the electronic edition alone. This published work and the nomenclatural acts it contains have been registered in ZooBank, the online registration system for the ICZN. The ZooBank LSIDs (Life Science Identifiers) can be resolved, and the associated information viewed through any standard web browser by appending the LSIDs to the prefix http://zoobank.org/. The LSIDs for this publication are: urn:lsid:zoobank.org:act:6CFF43A9-8C98-4027-A1DA-2838FE4D79F8; urn:lsid:zoobank.org:act:D67CCC72-E17D-450C-9193-231120527FDE; and urn:lsid:zoobank.org:act:3129A3DE-57E8-46CC-8036-86DC467EB056. The online version of this work is archived and available from the following digital repositories: PeerJ, PubMed Central SCIE, and CLOCKSS.

Results

Molecular analysis of genetic distances

The corrected p-distances under NJ of COI recovered six putative species (Fig. 1). Genetic distance analyses recovered groups corresponding to one putative new species (with bootstrap support value below 50%), the two previously described species N. pulex and N. xerophila (with high bootstrap support value, 89%), and three new species described herein (with high bootstrap support value, 98%). Bootstrap support values for all species were high (>89%) (Fig. 1). The average genetic p-interspecific distances of Naphrys species was 14% (min: 11%, max: 18.1%) (Table 2). Average interspecific p-distance between previously known species (N. pulex and N. xerophila) was 11.8%. Between new species (N. echeri sp. nov., N. tecoxquin sp. nov., and N. tuuca sp. nov.) and previously known species, higher interspecific average p-distances were observed, between 12.9% and 14%. With average values above 15.1%, Naphrys sp. had the highest average interspecific p-distance. For most species, intraspecific distances were below 1.61%, except for Naphrys sp. that showed a higher value (Table 3).

Figure 1 Neighbor-joining (NJ) with corrected p-distances tree constructed with COI sequences from different species of Naphrys.

Colors indicate putative species. Red numbers above branches represent significant bootstrap support values (<50%).

Table 2 Average genetic distances (p-distances) of COI among Naphrys species.

	1	2	3	4	5	
1. Naphrys pulex USA	-					
2. Naphrys xerophila USA	11.8	–				
3. Naphrys sp. USA	15.1	16.4	–			
4. Naphrys tecoxquinsp. nov. MEX	14.0	12.9	18.1	–		
5. Naphrys echerisp. nov. MEX	13.4	13.4	17.8	11.2	–	
6. Naphrys tuucasp. nov. MEX	13.0	13.6	17.2	11.0	11.1	

Table 3 Average genetic distance (p-distances) of COI within Naphrys species.

Specie	Distance	Standard error	
Naphrys pulex USA	1.61	0.26	
Naphrys xerophila USA	–	–	
Naphrys sp. USA	10.94	1.18	
Naphrys tecoxquinsp. nov. MEX	0	0	
Naphrys echerisp. nov. MEX	0.32	0.15	
Naphrys tuucasp. nov. MEX	0.34	0.19	

Molecular methods for species delimitation

The ASAP delimitation analysis recovered all six species (N. echeri sp. nov., N. tecoxquin sp. nov., N. tuuca sp. nov., Naphrys sp., Naphrys pulex, and Naphrys xerophila) with high (>93%) bootstrap support value (Fig. 2) from the NJ tree. GMYC and mPTP methods recovered the three new species described herein and one putative new species, while N. pulex was not recovered as one species (Fig. 2). The most incongruent result was observed in bPTP, which delimited 42 and 50 putative species under ML and IB variants, respectively. Only N. tecoxquin sp. nov. and N. xerophila were recovered by the ML variant of bPTP.

Figure 2 Maximum likelihood (ML) tree of Naphrys constructed with COI.

Colors represent putative species. Columns represent the different species delimitation methods. Numbers above branches represent Bootstrap support values for ML (>50% significant). Column abbreviations: Neighbor-joining (NJ); General Mixed Yule Coalescent (GMYC); Bayesian Poisson Tree Processes (bPTP) with maximum likelihood (ML) and Bayesian inference (IB) variants; Multi-rate Poisson Tree Processes (mPTP). Red numbers above branches represent bootstrap support values for ML (>50% significant).

Only N. xerophila was recovered under all methods and supported by a high bootstrap value (93%). Naphrys pulex shows the most incongruent results in all species delimitation methods, recovering 10 species in mPTP, 16 in GMYC, and 42 and 50 species in the ML and BI variants of bPTP method, respectively (Fig. 2). Nevertheless, N. pulex presents low intraspecific genetic distance (<2%) and high bootstrap support value (100%) (Table 3; Fig. 2).

Taxonomy

Family: Salticidae Blackwall, 1841	
Genus: Naphrys Edwards, 2003	
Tribe: Euophryini Simon, 1901	
Type species:HabrocestumacerbumPeckham & Peckham, 1909, by original designation (=Naphrys acerba).	

Emended diagnosis. After Richman (1981) and Edwards (2003). Naphrys species are characterized by their small size (2–6.1 mm) and dull, cryptic coloration (black and brown) (Figs. 3C–3D). Chelicera with one bicuspid promarginal tooth. Carapace high. First tibia has no more than two pairs of ventral macrosetae and leg III longer than leg IV (Tibia+Patella III >Tibia+Patella IV). Male palpal bulb is usually large with a proximal TL. Simple finger-like RTA and RSDL present. Also, palpal tibia with ventral apophysis (VTA). Embolar disk (ED) has a ventral conical projection. Embolus (E) possesses a J-shaped configuration, featuring a prolateral curvature in its distal part and an emerging projection in its proximal part (Figs. 4C–4H). Epigynum has a typical window structure with a median septum (Figs. 5C–5F). Copulatory openings (CO) are positioned along posterior, median (Figs. 5C–5F), or anteromedian edges of atria, with atrial rims intersecting them posteriorly. Rims fail to completely encircle the atria. Spermathecas (S) are nearly spherical, more or less contiguous medially, and half or more the diameter of the atria. They are positioned about halfway to entirely within the posterior part of atria as seen in ventral view (Figs. 5C–5F).

Figure 3 Naphrys acerba (Peckham & Peckham, 1909) from path to cable car, Cerro de la Silla, Guadalupe, Nuevo León, Mexico.

Red arrow indicates (A) Habitat, (B) Microhabitat. (C) and (D) Live female on leaf litter. Photos by Juan Maldonado-Carrizales.

Figure 4 Naphrys acerba (Peckham & Peckham, 1909).

Male habitus (A) dorsal and (B) ventral views Left palp (C) ventral, (D) retrolateral and (E) prolateral views. Drawings of left palp (F) ventral, (G) retrolateral and (H) prolateral views.

Figure 5 Naphrys acerba (Peckham & Peckham, 1909).

Female habitus (A) dorsal and (B) ventral views. Epigynum (C) dorsal and (D) ventral views. Drawings of epigynum (E) ventral and (F) dorsal views.

Current composition. Naphrys is composed of seven species: Naphrys acerba (Peckham & Peckham, 1909); Naphrys bufoides (Chamberlin & Ivie, 1944); Naphrys echeri sp. nov.; Naphrys pulex (Hentz, 1846); Naphrys tecoxquin sp. nov.; Naphrys tuuca sp. nov.; Naphrys xerophila (Richman, 1981).

Distribution. Canada, Mexico, and the United States.

Remarks. We emend the generic diagnosis based on copulatory organs of male and females.

Key to Naphrys species

1. Male ……………………………………………………………………………………2	
-. Female …………………………………………………………………………………8	
2. Dorsum of opisthosoma with two round, bright white spots (Fig. 4A) …………………... 3	
-. Dorsum of opisthosoma otherwise …………………………………………………….. 5	
3. Embolus thin and straight, larger than ED (Richman, 1981; Fig. 5). White setae covering all lateral side of prosoma (Edwards & Hill, 2008; Fig. 7) …………………………………. ……………..……………………………N. bufoides(Chamberlin & Ivie, 1944)	
-. Embolus thick and curve, shorter than ED (Richman, 1981; Figs. 8 and 16) …………...………………………………………………………………………………. 4	
4. Dorsum of the opisthosoma with a medial longitudinal white stripe that covers the anterior portion. Anterior part of prosoma exhibits bright, coppery bronze setae across surface (Metzner, 2024; Fig. 293) ………………………………. N. xerophila(Richman, 1981)	
-. Dorsum of the opisthosoma without medial longitudinal white stripe. Anterior part of prosoma is densely covered with a mixture of white, bronze, and black setae (Fig. 4A) …. ……………………………..……………………N. acerba(Peckham & Peckham, 1909)	
5. Dorsum of opisthosoma with an extended medial white longitudinal band that extends across the entire opisthosoma ………………………………………………………... 6	
-. Dorsum of opisthosoma otherwise………………………………………………….. 7	
6. Embolar disk (ED) bears a well-developed triangular process, next to the embolus, clearly visible in retrolateral view and smaller than embolus. Embolus thick and shorter than ED. Prosoma, in dorsal view, has white setae forming a V-shape mark, extending outwards from the sides of the PLE towards the pedicel ………………………………………N. echeri sp. nov.	
-. Embolar disk (ED) lacks a process. The thin embolus, larger than ED, folds at the midpoint, forming a gentle curve. Prosoma, in dorsal view, has white setae forming a Y-shape mark, extending outwards from the sides of the PLE ………………………………N. tuuca sp. nov.	
7. Embolus thick and curved, shorter than ED (Zhang & Maddison, 2015; Fig. 140). Anterior part of prosoma densely covered with a mixture of white and black setae (Edwards & Hill, 2008; Fig. 8) ………………………………………………………. N. pulex (Hentz, 1846)	
-. Embolus thick and straight. Anterior part of prosoma exhibits bright, coppery-bronze setae. Legs I–III dark brown color ………………………………….. N. tecoxquin sp. nov.	
8. Copulatory openings (CO) are located on the external lateral side of the S ………………………………………………………………………………………….. 9	
-. Copulatory openings (CO) located in different place …………………………………. 10	
9. Pyriform S. Light opisthosoma with four black spots in dorsal view, along with dark brown upwards chevron marks in the posterior last third …………………N. tecoxquin sp. nov.	
-. Circular S. Dark opisthosoma covered with coppery-bronze setae across surface and exhibiting mottled pattern of faint translucent markings (Fig. 6C) ………………N. echeri sp. nov.	
10. Copulatory ducts (CD) open into the epigynum forming transparent windows (W), with openings more than one-third the length of S (Figs. 5C–5F)……………………………. 11	
-. Copulatory ducts (CD) have circular opening, less than one-third the length of S. Copulatory openings (CO) located anteriorly to S (Richman, 1981; Fig. 18). Dark opisthosoma covered with brown and black setae across surface, with a longitudinal white stripe in the middle of the first third and a black chevron pattern on the remaining two-thirds (Metzner, 2024; Fig. 294) …………………………………N. xerophila(Richman, 1981)	
11. Dorsum of opisthosoma with two round, bright white spots (Figs. 3C, 3D, 5A) …………………………………………………………………………………………12	
-. Dorsum of opisthosoma otherwise……………………………………………………13	
12.- Copulatory openings (CO) located in center of epigynum, touching the anterior edge of S. Copulatory ducts (CD) have a unique loop, resembling a G-shape (Figs. 5C, 5E) …………………………………………N. acerba(Peckham & Peckham, 1909)	
-. Copulatory openings (CO) not touching anterior edge of S (Richman, 1981; Fig. 22) …………………………………………N. bufoides (Chamberlin & Ivie, 1944)	
13. Copulatory openings (CO) located in the middle of epigynum (Richman, 1981; Fig. 10) …..... ………………………………………………. N. pulex (Hentz, 1846)	
-. Copulatory openings (CO) located in the middle basal part of epigynum ………………………………..………........................................ N. tuuca sp. nov.	

Naphrys acerba (Peckham & Peckham, 1909)	
Figs. 3–5.	
Habrocestum acerbumPeckham & Peckham, 1909, p. 522, pl. 44, figs. 1–Ic.	

Holotype: Holotype not assigned by author. Syntypes: several males and one female from Travis County, Austin, Texas, USA, and one male from Georgia, USA. NOT EXAMINED. Naphrys acerba Edwards, 2003 p. 69, figs. 5–8 (Transferred from Habrocestum)

Other material examined. MEXICO: Nuevo León: six females (CAFBUM88003, CAFBUM88004, CABUM84234, CAFBUM84242, CAFBUM84256, CAFBUM84257), along path to cable car, Cerro de la Silla, Guadalupe municipality (lat. 25.655501, long. -100.254415, 587 m), oak forest, ground hand collecting, J. Maldonado Carrizales, F. Morales Martínez, E. G. Fuentes Ortiz cols., 21/X/2023. Tamaulipas: three males (CAFBUM88005) and three immatures (CAFBUM880040), Mr. Sabino’s ranch, highway Ciudad Victoria-Tula km 28 (lat. 23.606512, long. 99.229572, 1,473 m), oak forest, ground hand collecting, J. Maldonado Carrizales, F. Morales Martínez, E. G. Fuentes Ortiz cols., 20/X/2023.

Emended diagnosis. After Peckham & Peckham (1909) and Richman (1981). Naphrys acerba resembles N. bufoides and N. xerophila by possessing white, round spots on dorsal abdomen (Figs. 3C, 4A, 5A). However, it differs from N. xerophila by lacking a medial longitudinal white stripe covering anterior portion. Additionally, N. acerba can be distinguished from N. bufoides by its thicker embolus, which is shorter than ED (Figs. 4C, 4E, 4F, 4H). In females, CO of N. acerba are located centrally within the epigynum, touching anterior edge of S (Fig. 5C, 5E). This contrasts with N. bufoides, where CO do not reach anterior edge of S, and N. tuuca, where CO are positioned in middle basal part of epigynum.

Distribution. UNITED STATES: Texas; MEXICO: Coahuila, Nuevo León, Tamaulipas [Richman, 1981; WSC, 2024], Jalisco, Michoacán and Nayarit [present data].

Natural history. According to Richman (1981), this species appears to be associated with oak and juniper woodlands. Specimens used in this study were collected from upper leaf litter layer of oak forests (Quercus sp.) at 1,473 m in Tamaulipas, Mexico, within known range of the species. This also included disturbed areas into Monterrey City (Figs. 3A–3D).

Naphrys echeri sp. nov.	
Figs. 6–10.	
LSID: urn:lsid:zoobank.org:act:FFCFC48A-1827-4DCF-9096-DE8504E63251	

Type material: Male holotype, MEXICO: Michoacán, Cerro El Gigante, Jesús del Monte, Morelia (lat. 19.636605, long. -101.146877, 2,192 m), oak forest (Quercus sp.), ground hand collecting, J. Maldonado Carrizales, F. Morales Martínez, R. Cortés Santillán cols., 31/III/2023. (CARCIB-AR-047). Paratypes: one female (CARCIB-Ar-008), one male (CARCIB-Ar-0327) and one female (CARCIB-Ar-0328) with same collection data as for holotype. Jalisco: one male, one immature (CAFBUM84264) Piedras Bolas, Ahualulco de Mercado (lat. 20.653021, long. -104.057697, 1,907 m), oak forest (Quercus sp.), ground hand collecting, J. Maldonado Carrizales, G. L. López Solís, S. Montañez Hernández, N. Ruíz Hernández cols., 8/IV/2022. One female (CAFBUM88012) UMA Potrero de Mulas, San Sebastián del Oeste municipality (lat. 20.749852, long. -104.976763, 797 m) cloud forest, ground hand collecting, J. Maldonado Carrizales, E. G. Fuentes Ortiz cols., 13/XII/2022.

Other material examined. MEXICO: Jalisco: one female (CNAN-Ar011468) and one male (CNAN-Ar011467), beginning of the path to Cerro La Bufa, San Sebastián del Oeste municipality (lat. 20.758, long. -104.8438, 1,460 m), young pine forest, D. Guerrero, G. Contreras, C. Hutton, G.B. Edwards cols., 14/VI/2018. Three males, three immatures (CNAN-Ar011464), and one female (CNAN-Ar011462), Piedras Bolas, Ahualulco de Mercado municipality (lat. 20.64945, long. -104.05592, 1,863 m), oak forest (Quercus sp.), D. Guerrero, G. Contreras, C. Hutton and G.B. Edwards cols., 17/VI/2018.

Etymology. The species name “echeri” (/et eɾi/ native pronunciation) is a noun in apposition that means “land or soil” in the P’urépecha language, referring to the microhabitat where it inhabits. The P’urépecha state, which peaked in the 14th and 15th centuries before Spanish arrival, is known today as Michoacán, and represents the type locality of this species.

Diagnosis. Naphrys echeri sp. nov. resembles N. tuuca sp. nov. by males having an extended medial white longitudinal line on dorsal part of opisthosoma, which extends across the entire opisthosoma (Fig. 7A). However, N. echeri sp. nov. differs in possessing an ED that bears a well-developed triangular process (PED) next to embolus, clearly visible in retrolateral view (Figs. 7D, 7G). Naphrys echeri sp. nov. has a thick and straight E shorter than ED (Figs. 7C, 7E, 7F, 7h), whereas in N. pulex this is thick but curved, and in N. tuuca sp. nov. the E is thin and folds at midpoint forming a gentle curve, ultimately larger than ED. Naphrys echeri sp. nov. differs from N. tecoxquin sp. nov. and N. tuuca sp. nov. in morphology of its embolus apex, with N. echeri sp. nov. possessing a fine projection that abruptly narrows to a spine-like structure and is oriented towards the interior of the palp. Females of N. echeri sp. nov. share with N. tecoxquin sp. nov. the placement of CO on external lateral side of S, but differ in shape; in N. echeri sp. nov., S are circular (Figs. 8C–8F), while in N. tecoxquin sp. nov. they are pyriform.

Description. Male holotype (CARCIB-AR-047). Total length: 2.60. Prosoma 1.57 long and 1.22 wide. Darkish brown, with white setae forming a V-shaped mark, extending outwards from sides of PLE towards pedicel in cephalic region (Fig. 7A). Lower border covered with white seta forming a band. Ocular quadrangle (OQ) 0.30 long. Anterior eyes row (AER) 1.46 times wider than PER, AER 1.10 wide, PER 0.75 wide. Sternum reddish brown, 0.65 long, 0.50 wide. Labium reddish brown, as long as wide, 0.30 long, 0.30 wide. Endite 0.42 long, 0.17 wide, reddish brown, whitish anteriorly and square shaped (Fig. 7B). Opisthosoma 1.03 long and 0.95 wide; exhibiting a longitudinal band with white setae in dorsal view, covering more than half its width (Fig. 7A). Palp covered by white setae in dorsal view; in ventral view possesses a straight, short, and wide E that covers up to half distal part of cymbium (Figs. 7C, 7F, 9A, 10A). Ventral view of E with scales (Figs. 10A, 10C). A PED is present, easily seen in retrolateral view, triangular with fine projection on tip that abruptly narrows forming two spine-like structure (Figs. 7D, 7G, 9B, 10A–10C). Embolus apex and SP are oriented towards interior of palp (Figs. 9A, 10A–10B). Embolus apex presents one fine projection that abruptly narrows to a spine-like structure, while SP presents a multi-convex edge forming smooth ridges (Fig. 10D). Embolar disk (ED) completely rough and folded in anterior portion (Figs. 9A, 10A). Tegulum (T) yellow with darkish marks and wide RSDL occupying more than half of it, easily seen in retrolateral view (Figs. 7D, 7G). Furthermore, RSDL is divided in two, anterior loop is extremely curved forming a backwards “C” that extends from the middle of the T to its retrolateral edge. Posterior loop is curved anteriorly and straight in its most posterior part, forming a backwards “L” that does not touch retrolateral edge (Figs. 7D, 7G). Retrolateral tibial apophysis (RTA) wide at base, becoming smaller in distal part slightly anteriorly oriented (Figs. 7D, 7G, 9B, 9D). Ventral tibial apophysis (VTA) rounded with a large pit at the tip. It has faint lines running across its surface (Figs. 9A, 9C). Reddish brown legs with black bands. Legs I–II are pale with dark intersegmental markings, except for the joint between the metatarsus and tarsus. Legs III–IV exhibit dark intersegmental markings throughout. Leg formula 3412. Leg I 2.84 (0.90, 0.45, 0.60, 0.46, 0.42), Leg II 2.72 (0.92, 0.45, 0.52, 0.45, 0.38), Leg III 3.90 (1.20, 0.55, 0.82, 0.77, 0.47), Leg IV 3.80 (1.30, 0.50, 0.72, 0.82, 0.45).

Figure 6 Type locality of Naphrys echeri sp. nov. from Cerro El Gigante, Jesús del Monte, Morelia, Michoacán, Mexico.

Red arrow indicates (A) habitat and (B) microhabitat. (C) live female specimen in oak forest. Photos by Juan Maldonado-Carrizales.

Figure 7 Naphrys echeri sp. nov. male holotype (CARCIB-AR-047).

Habitus (A) dorsal and (B) ventral views Left palp (C) ventral, (D) retrolateral and (E) prolateral views. Drawings of left palp (F) ventral, (G) retrolateral and (H) prolateral views.

Figure 8 Naphrys echeri sp. nov. female allotype (CARCIB-Ar-008).

Habitus (A) dorsal and (B) ventral views. Epigynum (C) dorsal and (D) ventral views. Drawings of epigynum (E) ventral and (F) dorsal views.

Figure 9 Naphrys echeri sp. nov. male genitalia SEM micrographs.

Palp (A) prolateral and (B) retrolateral views. (C) Ventral tibial apophysis (VTA). (D) retrolateral tibial apophysis (RTA).

Female (CARCIB-Ar-008). Paler coloration, less pronounced than that of the male. Total length: 5.10. Prosoma 2.50 long and 1.90 wide. Darkish brown, with white and orange setae anteriorly (Fig. 8A). Lower border covered with white setae forming a band. Ocular quadrangle (OQ) 0.60 long. Anterior eyes row (AER) 1.27 times wider than PER, AER 1.40 wide, PER 1.10 wide. Sternum reddish brown with dark marks, 1.67 long, 0.87 wide. Labium black slightly longer than wide, 0.37 long, 0.32 wide. Endite 0.25 long, 0.65 wide, reddish brown, whitish anteriorly and ovoid shaped (Fig. 8B). Opisthosoma 2.60 long and 2.50 wide; covered with coppery bronze setae across surface and exhibiting mottled pattern of faint translucent markings (Fig. 8A). Epigynum slightly wider than long, 0.40 long, 0.34 wide. Copulatory openings (CO) located on external lateral sides of S. Circular S and a unique loop in CD forms a D-shape in each side of epigynum (Figs. 8C–8F). Median septum (MS) and sides have a smooth, trident-shaped with grooves or ridges on anterior part (Fig. 10E). Windows of epigynum (W) mostly smooth, but striated centrally (Fig. 10E). Reddish brown legs with black marks. Leg formula 3412. Leg I 3.72 (1.12, 0.70, 0.85, 0.65, 0.40), Leg II 3.67 (1.30, 0.62, 0.67, 0.62, 0.45), Leg III 5.52 (1.85, 0.80, 1.25, 1.00, 0.62), Leg IV 4.40 (1.57, 0.67, 1.12, 0.52, 0.50).

Distribution. MEXICO: Michoacán and Jalisco.

Natural history. The specimens were collected from leaf litter in oak forest (Quercus sp.) and cloud forest. Adults were mainly found from March to November (Figs. 6A–6C).

Naphrys tecoxquin sp. nov.	
Figs. 11–15.	
urn:lsid:zoobank.org:act:D67CCC72-E17D-450C-9193-231120527FDE	

Type material: Male holotype, MEXICO: Jalisco, Boca de Tomatlán, Cabo Corrientes (lat. 20.511861, long. -105.318, 36 m), tropical forest, ground hand collecting, J. Maldonado Carrizales, R. Cortés Santillán, E. G. Fuentes Ortiz cols., 13/IV/2023 (CARCIB-Ar-048). Paratypes: one female (CARCIB-Ar-009), one male (CARCIB-Ar-0329) and one female (CARCIB-Ar-0330) with same collection data as holotype; two males (CAFBUM84260-CAFBUM84261) and one female (CAFBUM84238) with same data as holotype.

Other material examined. MEXICO. Jalisco: one male (CAFBUM84232) and 12 immatures (CAFBUM84221), same collection data as holotype. 1 imm (CNAN-Ar011469), same collection data as paratype. One female (CNAN-Ar011471), Las Ánimas in same municipality as holotype (lat. 20.50002, long. -105.33869, 39 m), tropical forest, ground hand collecting, G. Contreras col., 6/IX/2018.

Etymology. The species name “tecoxquin” (/tek′o kin/ native pronunciation) is a noun in apposition in reference to the original native group that inhabited an extensive region covering the entire southern coast of Nayarit and neighboring coastal of Jalisco where type locality is found.

Diagnosis. Naphrys tecoxquin sp. nov. males possess bright, coppery bronze setae in the anterior part of the Prosoma (Figs. 11E, 12A), a light opisthosoma with four black spots in dorsal view, and upwards-pointing dark brown marks in posterior third (Fig. 12A). In contrast, N. echeri sp. nov. exhibits a dark opisthosoma covered with coppery bronze setae across its surface and displays a mottled pattern of faint translucent markings (Fig. 7A). Naphrys tecoxquin sp. nov. exhibits dark brown legs I–III (Fig. 11E), contrasting with the rest of species. Naphrys tecoxquin sp. nov. is similar to N. xerophila, but differs in having a thick and straight embolus (Fig. 12C–12H), in contrast to the curved embolus observed in N. xerophila and N. pulex. Naphrys tecoxquin sp. nov. differs from N. echeri sp. nov. and N. tuuca sp. nov. in morphology of its embolus apex, which is ventrally flat and dorsally convex, oriented towards the exterior of the pedipalp. The surface of the embolus apex in N. tecoxquin sp. nov. is sinuous with small projections. Additionally, N. tecoxquin sp. nov. lacks PED next to embolus, a characteristic of N. echeri sp. nov. (Figs. 7D, 7G, 9B, 10A–10C). In females of N. tecoxquin sp. nov., CO are located on external lateral side of S (Figs. 13C–13E). Naphrys tecoxquin sp. nov. differs to N. echeri sp. nov. in S shape, which is pyriform in N. tecoxquin sp. nov. (Figs. 13C–13F), but round in N. echeri sp. nov. (Figs. 8C–8F).

Figure 10 Naphrys echeri sp. nov. male genitalia SEM micrographs.

Embolus (A) ventral and (B) dorsal view. (C) Process on embolic disc (PED). (D) Sperm pore (SP) at embolus apex. (E) Female genitalia SEM micrograph epigynum ventral view.

Figure 11 Type locality of Naphrys tecoxquin sp. nov. from Boca de Tomatlán, Cabo Corrientes, Jalisco, Mexico.

Red arrow indicates (A) habitat and (B) microhabitat. (C) Red arrow indicates live specimen on floor. (D) Female live specimen and (E) male live specimen. Photos by Juan Maldonado-Carrizales.

Figure 12 Naphrys tecoxquin sp. nov. male holotype (CARCIB-Ar-048).

Habitus (A) dorsal and (B) ventral views. Left palp (C) ventral, (D) retrolateral and (E) prolateral views. Drawings of left palp (F) ventral, (G) retrolateral and (H) prolateral views.

Figure 13 Naphrys tecoxquin sp. nov. female allotype (CARCIB-Ar-009).

Habitus (A) dorsal and (B) ventral views. Epigynum (C) dorsal and (D) ventral views. Drawings of epigynum (E) ventral and (F) dorsal views.

Figure 14 Naphrys tecoxquin sp. nov. male genitalia SEM micrographs.

Palp (A) prolateral and (B) retrolateral views. (C) Ventral tibial apophysis (VTA). (D) Retrolateral tibial apophysis (RTA).

Figure 15 Naphrys tecoxquin sp. nov. male genitalia SEM micrographs.

Embolus (A) ventral view. (B) Sperm pore (SP) at embolus apex. (C) Female genitalia SEM micrograph epigynum ventral view.

Description. Male holotype (CARCIB-Ar-048). Total length: 2.90. Prosoma 1.74 long and 1.26 wide. Darkish brown, with white setae forming a U-shaped mark, extending outwards from sides of PLE towards pedicel, anterior part is covered by bronze setae (Fig. 12A). Lower border covered with white setae forming a band. Ocular quadrangle (OQ), 0.60 long. Anterior eye row (AER) 1.31 times wider than PER, AER 1.18 wide, PER 0.90 wide. Sternum dark with faint yellow marks, 0.62 long, 0.46 wide. Labium dark, wider than long, 0.15 long, 0.22 wide. Endite 0.27 long, 0.25 wide, reddish brown, whitish anteriorly, and square shaped (Fig. 12B). Opisthosoma 1.16 long and 0.92 wide, exhibiting two straight longitudinal bands forming a “V” that cover almost half of anterior opisthosoma. In central part, there is a black mark in shape of three triangles joined at base. Additionally, a white diamond-shaped mark is present in distal part (Fig. 12A). Palp covered by white setae in dorsal view; in ventral view, a thick and straight E covers up to half of distal part of the cymbium (Figs. 12C, 12F). Embolus apex and SP are oriented towards exterior of the palp (Figs. 14A, 15A). Embolus apex is ventrally flat and dorsally convex, oriented towards the exterior of pedipalp. Surface of the embolus apex is sinuous with small projections (Figs. 15A–15B). Embolar disk (ED) possesses a slight fold anteriorly, with striations at center (Figs. 14A, 15A). Tegulum (T) dark with faint yellow and orange marks. RSDL wide and easily seen in retrolateral view (Figs. 12D, 12G). Furthermore, RSDL is divided in two, anterior loop is gently curved similar to a closed parentheses “)”, extended on retrolateral edge. Adjacent, the posterior loop shares the same shape, but does not touch retrolateral edge (Figs. 12D, 12G). Retrolateral tibial apophysis (RTA) exhibits a densely striated surface along entire length. This apophysis projects in a straight orientation, gradually attenuating distally. Notably, RTA displays a slight dorsal orientation relative to the palp (Figs. 14B, 14D). Ventral tibial apophysis (VTA) is rounded and smooth (Figs. 14A, 14C). Leg I, femur, patella, tibia and metatarsus are dark brown with faint reddish-brown marks. Legs II–III, femur, patella and tibia are dark brown with faint reddish-brown marks, metatarsus amber, and tarsus yellow. Leg IV, femur, metatarsus, and tarsus yellow, patella and tibia dark brown with faint blackish-brown marks. Leg formula 3412. Leg I 2.81 (0.82, 0.48, 0.62, 0.45, 0.42); leg II 2.86 (0.85, 0.47, 0.60, 0.52, 0.41); leg III 3.83 (1.25, 0.47, 0.77, 0.81, 0.52); leg IV 3.75 (1.27. 0.58. 0.78. 0.57. 0.52).

Female (CARCIB-Ar-009). Paler coloration, less pronounced than that of the male, particularly on the prosoma. Total length: 2.68. Prosoma 1.50 long and 1.10 wide, darkish brown, with anterior part covered with black and orange setae (Fig. 13A); lower border covered with white setae forming a band. Ocular quadrangle (OQ), 0.70 long. Anterior eyes row (AER) 1.50 times wider than PER, AER 1.08 wide, PER 0.72 wide. Sternum reddish brown with dark marks, 0.62 long, 0.46 wide. Labium black, wider than long, 0.22 long, 0.46 wide. Endite 0.28 long, 0.24 wide, reddish brown, and ovoid shaped (Fig. 13B). Opisthosoma 1.18 long and 0.92 wide; light with four black spots in dorsal view, along with dark brown upwards chevron marks in posterior last third (Fig. 13A). Epigynum longer than wide, 0.82 long, 0.46 wide. Copulatory openings (CO) are located on external lateral sides of S. Pyriform S and a unique loop in CD forms a D-shape on each side of the epigynum (Figs. 12C–13F). Median septum (MS) and sides smooth, trident-shaped, with grooves on anterior edges of W (Fig. 15C). Windows of epigynum (W) longer than wide, mostly smooth, but striated at center (Fig. 15C). Reddish brown legs with black marks. Leg formula 3412. Leg I 2.25 (0.67, 0.45, 0.47, 0.37, 0.27); leg II 2.12 (0.55, 0.40, 0.50, 0.35, 0.32); leg III 3.27 (1.05, 0.45, 0.70, 0.60, 0.47); leg IV 3.10 (1.00, 0.40, 0.67, 0.65, 0.37).

Distribution. MEXICO: Jalisco.

Natural history. The specimens were collected from leaf litter in tropical dry forests with broad-leaved trees. Adults were mainly found from April to July and from September to November (Fig. 11).

Naphrys tuuca sp. nov.	
Figs. 16–22.	
LSID: urn:lsid:zoobank.org:act:3129A3DE-57E8-46CC-8036-86DC467EB056	
	

Type material: Male holotype, MEXICO: Nayarit, male from Cerro San Juan, Tepic (lat. 21.505877, long. -104.924464, 1,121 m), oak forest (Quercus sp.), ground hand collecting, J. Maldonado Carrizales, R. Cortés Santillán col., 24/V/2023 (CARCIB-Ar-049). Paratypes: one female (CARCIB-Ar-010), three males (CARCIB-Ar-0331; CAFBUM880039) and three females (CARCIB-Ar-0332; CAFBUM880021) with same collection data as holotype.

Other material examined. MEXICO. Nayarit: two males (CAFBUM880001; CAFBUM880002), one female (CAFBUM880075), same data as holotype. One male (CNAN-Ar011460), same data as holotype (CNAN-Ar011461). Three males and three females (CNAN-Ar011461), Ceboruco Volcano, Jala municipality (lat. 21.1149, long. -104.5014, 1,916 m), wet glen, D. Guerrero, G. Contreras, C. Hutton, and G.B. Edwards col., 16/V/2018.

Etymology. The species name “tuuca” (/ uuk’a/ native pronunciation) is a noun in apposition that means “spider” in the Wixárika language. Wixárika people are native to the Sierra Madre Occidental range in Nayarit state, where the type locality is found.

Diagnosis. Prosoma in dorsal view of N. tuuca sp. nov. has a unique characteristic white setae forming a Y-shaped mark, extending outwards from sides of PLE (Figs. 16C, 17A). In contrast, N. echeri sp. nov. exhibits white setae forming a V-shaped mark in this region (Fig. 7A). Naphrys tuuca sp. nov. has a dark opisthosoma covered with coppery-bronze setae across surface (Figs. 16C, 17A), similar to N. echeri sp. nov.; nevertheless, N. tuuca sp. nov. has a distinct mottled pattern of white markings and a medial longitudinal smooth white stripe that covers anterior portion of the opisthosoma (Figs. 16C, 17A). Males of N. tuuca sp. nov. possess a thin embolus (Figs. 17C–17H). Embolus is larger than ED and folds at midpoint, forming a gentle curve (Figs. 17E, 17H), in contrast to thin and straight embolus observed in N. bufoides. Similar to Naphrys tecoxquin sp. nov., embolus apex of N. tuuca sp. nov. is curved and oriented towards the exterior of palp. Surface of embolus apex in N. tuuca sp. nov. is smooth. Additionally, N. tuuca sp. nov. lacks PED, which is present in N. echeri sp. nov. Females of N. tuuca sp. nov. present CO located in middle basal part of epigynum (Fig. 18C, 18E), differing from central location of CO observed in N. acerba, N. bufoides and N. pulex.

Figure 16 Type locality of Naphrys tuuca sp. nov. from Cerro San Juan, Tepic, Nayarit, Mexico.

Red arrow indicates (A) habitat and (B) microhabitat (C) live male specimen (D) live female eating a Collembola in field. (E) Live female eating a larva of Drosophila melanogaster Meigen, 1830 in captivity. Photos by Juan Maldonado-Carrizales.

Figure 17 Naphrys tuuca sp. nov. male holotype (CARCIB-Ar-049).

Habitus (A) dorsal and (B) ventral views. Left palp (C) ventral, (D) retrolateral and (E) prolateral views. Drawings of left palp (F) ventral, (G) retrolateral and (H) prolateral views.

Figure 18 Naphrys tuuca sp. nov. female allotype (CARCIB-Ar-010).

Habitus (A) dorsal and (B) ventral views. Epigynum (C) dorsal and (D) ventral views. Drawings of epigynum (E) ventral and (F) dorsal views.

Figure 19 Naphrys tuuca sp. nov. male genitalia SEM micrographs.

Palp (A) prolateral and (B) retrolateral views. (C) Ventral tibial apophysis (VTA). (D) Retrolateral tibial apophysis (RTA).

Figure 20 Naphrys tuuca sp. nov. male genitalia SEM micrographs.

Embolus (A) ventral view. (B) Sperm pore (SP) at embolus apex. (C) Female genitalia SEM micrograph epigynum ventral view.

Figure 21 Retrolateral view of male Naphrys legs.

Left column indicates leg number. Top row indicates species. (A, E, I & M) legs I, II, III & IV of Naphrys acerba, respectively; (B, F, J & N) legs I, II, III, IV of Naphrys echeri sp. nov., respectively; (C, G, K & O) legs I, II, III, IV of Naphrys tecoxquin sp. nov., respectively; (D, H, L & P) legs I, II, III, IV of Naphrys tuuca sp. nov., respectively.

Figure 22 Known distribution records of the Mexican species of Naphrys.

Star: N. acerba. Diamond: N. echeri sp. nov.. Circle: N. tecoxquin sp. nov. Cross: N. tuuca sp. nov. Colors represent the biogeographical provinces following Escalante, Rodríguez-Tapia & Morrone (2021). Blue: Transmexican Volcanic Belt province. Green: Sierra Madre del Sur Province. Pink: Sierra Madre Oriental. Yellow: Pacific Lowlands.

Description. Male holotype (CARCIB-Ar-049). Total length: 2.48. Prosoma 1.42 long and 1.10 wide, dark with white setae forming a Y-shaped mark, extending outwards from sides of PLE towards pedicel (Figs. 16C, 17A). Lower border covered with white setae forming a band. Ocular quadrangle (OQ), 0.74 long. Anterior eye row (AER) 1.53 times wider than PER, AER 0.98 wide, PER 0.64 wide. Sternum dark with faint amber marks, 0.72 long, 0.50 wide. Labium dark, wider than long, 0.17 long, 0.25 wide. Endite 0.35 long, 0.27 wide, amber, and square-shaped (Fig. 17B). Opisthosoma 1.06 long and 0.88 wide, exhibiting a longitudinal band with white setae in dorsal view, covering one third of width (Figs. 16C, 17A). Palp covered by white setae in dorsal view, with a thin embolus in ventral view, larger than ED, which folds at midpoint, forming a gentle curve (Figs. 17C, 17E, 17F, 17H, 19A, 20A). Embolus apex exhibits a lateral flattening, resulting in a dorsally convex shape; oriented outwards from the main body of the palp. Embolus apex surface with smooth contours (Figs. 20A–20B). Embolar disk (ED) exhibits unfolded anterior margin, and central region displays a higher concentration of striations (Figs. 19A, 20A). Tegulum (T) dark with faint yellow and orange marks, RSDL wide, easily seen in retrolateral view (Figs. 17D, 17G). Furthermore, RSDL is divided in two, with anterior loop extremely curved, forming a backwards “C” that extends from middle of T to its retrolateral edge. Posterior loop is curved anteriorly and straight in its most posterior part, forming a hooked-shape that does not touch retrolateral edge (Figs. 17D, 17G). Retrolateral tibial apophysis (RTA) exhibits sparse striations along its entire length. This structure projects in a straight orientation, gradually attenuating distally and displaying a slight anterior orientation (Figs. 17D, 17G, 19B, 19D). Ventral tibial apophysis (VTA) presents a conical structure with a roughened surface texture and a small notch distally (Figs. 19A, 19C). Legs I–II are pale with dark intersegmental markings, except for the joint between the metatarsus and tarsus. Legs III–IV exhibit dark intersegmental markings throughout (Figs. 21D, 21H, 21L, 21P). Leg formula 3412. Leg I 2.71 (0.78, 0.47, 0.50, 0.49, 0.45); leg II 2.68 (0.96, 0.45, 0.51, 0.50, 0.24); leg III 3.91 (1.26, 0.65, 0.78, 0.74, 0.47); leg IV 3.60 (1.10, 0.45, 0.76, 0.87, 0.49).

Female (CARCIB-r-010). Paler coloration, less pronounced than that of the male. Total length: 3.64. Prosoma 1.64 long and 1.34 wide, darkish brown, anterior part covered with white and orange setae (Figs. 16D, 16E, 18A). Lower border covered with white setae forming a band. Ocular quadrangle (OQ) 0.68 long. Anterior eyes row (AER) 1.47 times wider than PER, AER 1.18 wide, PER 0.80 wide. Sternum reddish brown with dark marks, 0.67 long, 0.57 wide. Labium dark with faint amber marks, wider than long, 0.20 long, 0.27 wide. Endite 0.37 long, 0.25 wide, reddish brown, and ovoid shaped (Fig. 18B). Opisthosoma 2.00 long and 1.80 wide, dark, covered with coppery-bronze setae across surface, with a mottled pattern of white markings and a medial longitudinal smooth white stripe covering anterior portion (Fig. 18A). Epigynum slightly wider than long, 0.30 long, 0.34 wide. Copulatory openings (CO) located in middle basal part of epigynum. Circular S and a unique loop in CD form a D-shape in each side of epigynum (Figs. 18C–3F). Median septum (MS) exhibits a smooth surface texture, while anterior edges of W present grooves (Fig. 20C). Overall surface of W exhibits a slightly roughened texture. Windows of epigynum (W) as long as wide (Fig. 20C). Legs yellow with dark marks, metatarsus amber, and tarsus yellow. Legs III and IV yellow with dark bands near segment junctions. Leg formula 3412. Leg I 2.82 (0.88, 0.52, 0.56, 0.50, 0.36); leg II 2.86 (0.92, 0.44, 0.58, 0.58, 0.34); leg III 4.14 (1.34, 0.58, 0.82, 0.90, 0.50); leg IV 4.06 (1.30, 0.56, 0.80, 0.82, 0.58).

Distribution. MEXICO: Nayarit.

Natural history. The specimens were collected from leaf litter in oak forests (Quercus sp.). Adults were mainly found from May to September. This species was observed to prey on Collembola (Fig. 16D).

Discussion

Species delimitation within the family Salticidae has increasingly relied on a combination of molecular and morphological data. This trend is evident in studies that employ a phylogenetic perspective (Maddison, 2016a; Maddison, 2016b; Cala-Riquelme, Bustamante & Salgado, 2022; Maddison et al., 2022). While genomic data can also be a reliable approach (Girard et al., 2021; Lin, Yang & Zhang, 2024), it typically requires greater resource investments and analysis time. In contrast, studies integrating diverse data sources for species delimitation within Salticidae remain relatively scarce.

A notable example is the work by Trębicki et al. (2021), who addressed taxonomic ambiguities within the genus Cytaea Keyserling, 1882 and related species. The authors attributed this taxonomic confusion to poor original diagnoses and descriptions within the genus. To resolve this issue, Trębicki et al. (2021) employed a combined approach, analyzing both the morphology of the holotype specimens and utilizing the Automatic Barcode Gap Detection (ABGD) method based on a NJ tree constructed with COI gene sequences. Their results revealed that previously recognized “similar species” were synonymous with the Cytaea holotype, prompting the authors to formally synonymize these taxa.

While the authors employed a distance-based delimitation method (NJ tree) to clarify the identity of ambiguous species, in our work we take a more comprehensive approach, incorporating tree-based molecular analyses. To avoid future confusion, we also present an emended diagnosis of the genus Naphrys. These comprehensive resources aim to facilitate accurate species and genus-level determinations.

Bopearachchi, Eberle & Benjamin (2022) further exemplify the application of molecular methods for species delimitation within Salticidae. Their study aimed to clarify the species diversity within the genus Ballus C. L. Koch, 1850 in Sri Lanka. Three species had been previously reported for this region, described in the late 19th and early 20th centuries. To address this taxonomic uncertainty, Bopearachchi, Eberle & Benjamin (2022) employed a multifaceted approach, integrating morphological data with sequence data from three genes (COI, H3, 28S). They utilized multiple species delimitation methods, including ABGD, mPTP, and Bayesian Multi-Locus Species Delimitation (BPP). Notably, all applied methods yielded congruent results, indicating that the three previously recognized Ballus species represented a single species with consistent morphological characteristics and no significant genetic differentiation.

Similar to our work, the authors employed multiple molecular methods to investigate species diversity within a genus containing previously described species. In our study, the mPTP method, also used by Bopearachchi, Eberle & Benjamin (2022), not only confirmed the identity of the previously known species N. xeophila, but also supported the designation of three new species.

Finally, Phung et al. (2024) employed a combined approach for species delimitation within the genus Phintella Strand, 1906 and related Phintella-like spiders. Their approach utilized three distinct methods: one distance-based method (ASAP) and two tree-based methods (Bayesian version of GMYC and BPP). These methods were used to delineate putative new species based on available genetic data. Furthermore, the authors recognized the challenge of strong sexual dimorphism within Phintella. They addressed this limitation by incorporating the same methods to assign male–female combinations for approximately one-third of the species where such pairings were unknown. The analyses by Phung et al. (2024) resulted in the identification of 22 distinct species, with 11 potentially representing undescribed taxa. Nevertheless, it is important to note that the study did not formally establish new species through the nomenclatural act.

Concordant with our findings, Phung et al. (2024) applied various methods for species delimitation. The distance-based ASAP method yielded a lower species count similar to our results. Conversely, tree-based methods (bGMYC and BPP) led to overestimations, as we also observed. Both studies endorse the utility of the COI gene for preliminary detection of potentially undescribed species, which subsequently have to be described as performed in this work.

Similar to the challenges encountered in the previous discussed studies, the Euophryini tribe exhibits numerous taxonomic uncertainties. These difficulties often stem from poor original species descriptions, limited knowledge of sexual dimorphism (e.g., only one sex known for some species), and high morphological similarity among species. To overcome these limitations, researchers have increasingly employed a combination of multiple methods (e.g., morphological and molecular data) for species delimitation (Navarro-Rodríguez & Valdez-Mondragón, 2020; Candia-Ramírez & Francke, 2021; Cala-Riquelme, Bustamante & Salgado, 2022).

Morphological characters, particularly sexual characteristics, remain indispensable for robust species diagnosis, identification, and delimitation (Valdez-Mondragón, 2020). This is due, in part, to the typically low level of intraspecific variation and high level of interspecific variation observed in spider genitalia (Eberhard, 1985; Eberhard et al., 1998), making this characteristic a valuable diagnostic tool (Valdez-Mondragón, 2013; Valdez-Mondragón, 2020; Valdez-Mondragón & Francke, 2015). In our study, we delimited different species through morphological characters, some of which were particularly diagnostic. For instance, the presence of a clearly visible PED in N. echeri sp. nov. and the distinctive shape of S readily distinguished this species from its congeners.

Modern taxonomic practices increasingly emphasize the integration of multiple data sources for species validation and delimitation. This combined approach strengthens the evidence for species boundaries and provides a more comprehensive understanding of the newly described taxa. In this way, the study herein represents the first where new species are described within the Salticidae family through species delimitation methods based on molecular data (both distance and tree-based).

Compared to other genes, the use of the COI gene has proven to be an effective tool for species delimitation in spiders (Trębicki et al., 2021; Valdez-Mondragón et al., 2019; Navarro-Rodríguez & Valdez-Mondragón, 2020; Nolasco & Valdez-Mondragón, 2022; Phung et al., 2024). Naseem & Muhamman (2016) identified Salticidae in citrus orchards using the COI gene with interspecific values of nucleotide divergence between 9.96–11.91%. Yamasaki et al. (2018) found higher interspecific values of nucleotide divergence (14.1–18.2%) in their redescription of the genus Chrysilla, based on morphology and DNA barcoding. Those studies serve as a reference for variation among different species. The interspecific genetic divergences found in this work were greater than 11% (mean: 14%, min: 11%, max: 18.1%), fitting within the range previously reported for Salticidae.

For many taxonomic groups, a 3% genetic divergence threshold is often used to define species boundaries (Sbordoni, 2010). However, this value can vary across animal groups and even among closely related species due to differences in evolutionary rates (Trębicki et al., 2021). Previous studies (Vink, Dupérré & McQuillan, 2011; Richardson & Gunter, 2012; Blagoev et al., 2016; Trębicki et al., 2021) have reported a broad range of intraspecific genetic divergences within the Salticidae family, ranging from less than 0.5% to 7.57%.

Our results (Table 3) fit within this established range for Salticidae, except for Naphrys sp., which exhibited a higher divergence value of 10.94%. Unfortunately, we were unable to examinated morphologically the specimens. Our identification of this taxon was solely based on genetic data retrieved from GenBank. To conclusively determine the diagnostic characters of this species, morphological analysis is indispensable. Of note is Naphrys pulex, which despite inconsistencies in some species delimitation methods, showed observed intraspecific variation less than 2%, which falls well within the expected range for species of Salticidae.

Among the methods tested in this work, ASAP recovered the lowest number of species, similar to the findings by Phung et al. (2024) with Salticidae. Guo & Kong (2022) suggested that the distance-based approach is generally superior to the tree-based approach, with the ASAP method being the most efficient. As in Phung et al. (2024), our use of GMYC, bPTP, and mPTP methods resulted in a significantly higher number of delineated species. This contrast to previous studies with other groups (Mygalomorphae and Araneomorphae) of spiders (Ortiz & Francke, 2016; Valdez-Mondragón et al., 2019; Navarro-Rodríguez & Valdez-Mondragón, 2020), in which a lower number of species were typically identified using similar methods. This discrepancy might be attributed to the limitations of GMYC and PTP methods. As discussed by Luo et al. (2018) and Guo & Kong (2022), these methods can be particularly sensitive to gene flow, which can disrupt the clear correlation between population size and divergence time, potentially leading to an overestimation of species boundaries. This overestimation issue could explain the differences found in the tree-based methods of the molecular analysis for N. pulex, despite the low genetic intraspecific distances observed (<2%).

Hamilton, Formanowicz & Bond (2011) emphasized the utility of geographical data in species delimitation. In our study, the different Naphrys species present in Mexico can be separated by their distribution. Naphrys pulex is widespread throughout the biogeographic Alleghany subregion corresponding to eastern Canada and the United States (Escalante, Rodríguez-Tapia & Morrone, 2021). Naphrys xerophila is distributed only in the southeastern coastal plains of the United States through the Austroriparian biogeographic province within the Alleghany subregion (Richman, Cutler & Hill, 2012; Escalante, Rodríguez-Tapia & Morrone, 2021). Their distribution is limited by the increased aridity in the western and southern boundaries of the Alleghany subregion (Takhtajan, 1986; Escalante, Rodríguez-Tapia & Morrone, 2021).

Prior to this study, the only known species present in Mexico was N. acerba, which is distributed in the northern part of the Sierra Madre Oriental biogeographical province in the northeast of the country. Naphrys tecoxquin sp. nov. inhabits a distinct biogeographical province, the Pacific Lowlands. This province corresponds to a narrow, uninterrupted strip along the Pacific coast (Morrone, 2019). Naphrys tuuca sp. nov. and N. echeri sp. nov. are distributed within the Trans-Mexican Volcanic Belt (TVB) province. This province corresponds to the set of volcano mountain ranges that traverses the country from west to east (Morrone, 2019).

Within the TVB, N. tuuca sp. nov. inhabits the western mountain zone. In contrast, N. echeri sp. nov. occupies the central mountains of the TVB (Fig. 22). Naphrys echeri sp. nov. also occurs in the eastern mountains of Mexico, specifically in the northern part of the Sierra Madre del Sur (SMS) province, a mountain system that runs in parallel to the Pacific Ocean coast in a northwest-southeast direction. Nevertheless, its continuity is interrupted by a series of valleys, with rivers typically flowing above 1,000 m (Hernández-Cerda, Azpra-Romero & Aguilar-Zamora, 2016; Morrone, 2019). The SMS and TVB provinces are both part of the Mexican Transition Zone (MTZ). The MTZ exhibits a unique combination of characteristics that distinguish it from other transition zones. Notably, it harbors a remarkable mixture of Nearctic and Neotropical taxa.

Geographical barriers play a key role in the differential distribution of N. echeri sp. nov. and N. tuuca sp. nov. The SMS mountain range breaks through a tectonic graben of volcanic plateaus, with stratovolcanoes developing along its margins such as the Ceboruco Volcano (Blanco y Correa, Pérez & Cruz-Medina, 2021). The easternmost locality for N. tuuca sp. nov. is separated from western localities of N. echeri sp. nov. (Piedras Bolas in the TVB and Potrero de Mulas in the SMS) by extensive alluvial plains (up to 25 km wide) and deep clefts formed by the Ameca River (Valdivia-Ornela & Castillo-Aja, 2001; Correa, Pérez & Cruz-Medina, 2021; Valero-Padilla, Rodríguez-Reynaga & Cruz-Angón, 2017).

The species described herein are the southernmost representatives of the genus. Contrary to prior assumptions by Edwards (2003) that the genus has a Nearctic distribution, our findings reveal the presence of these species in the Neotropical region, suggesting a broader geographical range. While the present work focused on western Mexico, further exploration particularly in the south is likely to yield additional undescribed species. This study also provides the first precise locality data for N. acerba within Mexico, previously known only from historical records.

Our study demonstrates the utility of the COI gene for robust species-level delimitation within the Naphrys genus. This finding is supported by the high congruence observed among most methods employed. Additionally, morphological characters, particularly the male palps and female epigynes, proved to be reliable features for the identification and diagnosis of Naphrys species.

Supplemental Information

Supplemental Information 1 Sequences cut and aligned

The first author thanks the Programa Institucional de Doctorado en Ciencias Biológicas (PIDCB), Facultad de Biología, Universidad Michoacana de San Nicolás de Hidalgo (UMSNH), for the academic training during the research, as well as to the PhD synodal committee for their valuable contributions, comments, and suggestions in the realization of this work. The authors appreciate the support of technician Ariel Arturo Cruz Villacorta in the operation of the scanning electron microscope (SEM) in the Electron Microscopy Laboratory at Centro de Investigaciones Biológicas del Noroeste, S.C.

Additional Information and Declarations

Competing Interests

Author Contributions

Field Study Permissions

DNA Deposition

Data Availability

New Species Registration

The authors declare there are no competing interests.

Juan Maldonado-Carrizales conceived and designed the experiments, performed the experiments, analyzed the data, prepared figures and/or tables, authored or reviewed drafts of the article, and approved the final draft.

Alejandro Valdez-Mondragón conceived and designed the experiments, analyzed the data, prepared figures and/or tables, authored or reviewed drafts of the article, and approved the final draft.

María L. Jiménez-Jiménez conceived and designed the experiments, analyzed the data, authored or reviewed drafts of the article, and approved the final draft.

Javier Ponce-Saavedra conceived and designed the experiments, analyzed the data, authored or reviewed drafts of the article, and approved the final draft.

The following information was supplied relating to field study approvals (i.e., approving body and any reference numbers):

Field collections were approved by Faculty of Biology, Universidad Michoacana de San Nicolás de Hidalgo.

The following information was supplied regarding the deposition of DNA sequences:

The genes are available at GenBank:

- Naphrys echeri: PP123908; PP123905; PP123902; PP123909; PP123903; PP123900; PP123901.

- Naphrys tecoxquin: PP123899; PP123898; PP123906.

- Naphrys tuuca: PP123904; PP123910; PP123907.

The following information was supplied regarding data availability:

The raw data is available in the Supplemental Files.

The following information was supplied regarding the registration of a newly described species:

Publication LSID: urn:lsid:zoobank.org:pub:BD59F63E-0BA4-4C72-AA28-01E3E46633F7.

Naphrys echeri LSID: urn:lsid:zoobank.org:act:6CFF43A9-8C98-4027-A1DA-2838FE4D79F8.

Naphrys tecoxquin LSID: urn:lsid:zoobank.org:act:D67CCC72-E17D-450C-9193-231120527FDE.

Naphrys tuuca LSID: urn:lsid:zoobank.org:act:3129A3DE-57E8-46CC-8036-86DC467EB056.

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
