# Peer review of "Three new species of the spider genus Naphrys Edwards (Araneae, Salticidae) under morphology and molecular data with notes in the distribution of Naphrys acerba (Peckham & Peckham) from Mexico"

_PeerJ, doi:10.7717/peerj.18775_

## Round 0.1 · original submission · Major Revisions

· Academic Editor

Major Revisions

The authors need to include all suggestions given by the reviewers, who are largely positive about your research.

·

Basic reporting

The manuscript brings new information about the most diverse family of spiders. In addition, the use of different methods in species delimitation makes the manuscript even more efficient and described through integrative taxonomy. Authors should pay attention to taxomic lists, adjustments in taxonomic descriptions, standardization of type material and revision of figures. The figures are saved in PNG, which makes the background transparent. Many are blurred and have a non-standard background. Review mainly the habitus, illustrations and SEM. Please provide a legend of the symbols of the species in distribution on the map figure. In addition, some adjustments in paragraph ordering. As soon as the adjustments and corrections are made, the work will be ready for acceptance and publication.

Experimental design

no comment

Validity of the findings

no comment

Additional comments

Were all authors discoverers of the new species? Please check this for all new species. In case of disagreement, please insert the authors' surnames in order of discovery. If not, all species are given the long name exemple: Naphrys echeri Maldonado-Carrizales, Valdez-Mondragón, Jiménez-Jiménez & Ponce-Saavedra sp. nov.

·

Basic reporting

Dear Academic Editor,
Dear Authors

This is a VERY important manuscript.

I really liked the idea: describing new species for the genus Naphrys, together with generating detailed morphological analysis and molecular based species delimitation. Integrative taxonomy, whether possible, should be a standard way for introducing new taxa for science, but in the spider, taxonomy is still not very common, thus such articles should be considered as valuable. Taxonomy itself is difficult science and requires much knowledge about the studied group, and many additional skills. Describing new taxa under integrative taxonomy is crucial because it decreases the subjective way of introducing new taxa to science, while some of them may become an interesting model for future biological studies. Taxonomic works that use both data on the morphology of the spiders studied, ultra structures and DNA sequences, as the authors mention, are still rare. I think the authors did a good job planning the research.
Starting from the title and abstract, the proposed manuscript sounds interesting and important.
I'm not a native English speaker, but I found the manuscript written in clear, unambiguous, technically correct language. An article conforms to professional standards of courtesy and expression.

I'm satisfied that I can provide a review of the proposed manuscript, but as a reviewer I must look critically on the general picture and on some details as well, to help authors to improve and polish manuscript to be ready for publication in the PeerJ journal.

Experimental design

There are a few problems with the manuscript though, which should be addressed.

Introduction:

paragraph 62-66 rows should be moved up between 46-47 rows, this will continue the consistency of the text. This change may make introductions easier to follow.

For species delimitation, authors have used one gene fragment - COI, common for the DNA barcoding - standardized by Hebert et al. in 2003 - it's worthy to mention and to cite in the introduction.

Materials & Methods

Taxon sampling (150-151) The molecular analyses were carried out with a total of 110 specimens, including one undescribed species of Naphrys and three new Naphrys species described herein. - It's unclear how many specimens were collected for this study, and how many are from repositories (GeneBank, BOLD).

DNA extraction, amplification, and sequencing:

(157-159) The DNA was isolated from legs using proteinase K/phenol/chloroform following the protocol by Hillis et al. (1996). Briefly, at least 1 μL of tissue was incubated at 60°C for 24 hours with a digestion buffer

From how many specimens the DNA were isolated, amplified and sequenced for this study? From how many legs the DNA was isolated - one, two or more?

"Briefly, at least 1 μL of tissue was incubated" using a microliter to the tissue is incorrect - you can use a milligram, but for this - each spider leg should be dried and weighed. It's not necessary to provide such information - genomic DNA was isolated from 1 or 2 legs - that's enough.

I would propose to delete all detailed descriptions for DNA isolation. Phenol/Chloroform method for DNA isolation is common, cost efficient, accurate, but also time consuming - if protocol from cited Hillis et al. (1996) was not significantly modified - there's no need for detailed description. I may suggest that incubation in the temperature of 60 Celsius might be too high - incubation in about 55-56 Celsius is the most recommended. Incubation of 24 hours is in general much too long - for relatively fresh specimens, collected months up to a few years before the isolation - from a few up to 12 hours of incubation is enough, longer might cause DNA degradation. Long lysis (up to 3 days) is rarely applied for some difficult cases as old museum specimens, but still questionable.
PCR reaction conditions (173-180) were following Folmer et al (1994)? or were modified? Annealing temperature is quite high and 35 cycles is too much (high possibility of nonspecific PCR products).

(180-181) - Expected PCR product (amplification with LCO/HCO primer pair), has expected size of 650 base pairs, thus, to check its length on the gel the ladder should have size up to 1000 bp - not 100bp as in the text (I believe that it’s just missing 0).

(185-186) - Sequencing was carried out in Psomagen, Maryland, United States. Do PCR products were purified in Psomagen or in the CMEB? Please precise on which sequencing platform the sequencing was carried out? Was it the Sanger platform? If yes - each of the purified PCR products was sequenced in one (LCO) or both directions (LCO/HCO)? These questions are related to the final results of sequencing - COI gene fragments amplified with standard LCO/HCO gene fragments should have about 650 base pairs while sequences obtained for this study are about 589 base pairs. Why did that happen? Must be clarified.

(188-192) Sequence editing and alignment:

Lack of crucial information about the sequence alignment. How many COI sequences were obtained for this study - here must be provided GeneBank accession numbers or table with the list of species/specimens and accession numbers from Genbank or BOLD - if sequences used for this study were already published, then citations should be provided (for some sequences used in this study, I found already published articles (for example: https://doi.org/10.1111/icad.12697 or https://doi.org/10.1139/z05-024).
Having final alignment of 110 COI sequences used for this study, authors should describe its characteristics - Length, number of nucleotides, transition to transversion ratio etc. Before the final analysis, an alignment should be checked for presence/absence of stop codons - I checked an alignment for these issues - it's totally correct (no stop codons). Most of the sequences already published, and used in this study, have size over 600 base pairs - why was an alignment trimmed to 589 bp? It's not wrong, but should be also explained. As far as I understand, authors trimmed the whole alignment to the size of 13 sequences obtained for this study (589 bp) - this was not necessary, while within further analysed alignment contained shorter sequences too. In further analysis authors were using "Pairwise deletion" correct for analysis of unequal (in length) sequences.

In supplementary files you provided the table with a list of sequences and an alignment – here you should refer to it.

Molecular analysis and species delimitation:

There is no need to explain details of each delimitation method - for example:

Assemble Species by Automatic Partitioning (ASAP) - delete rows 210-119, leave ASAP (Puillandre et al. 2021), analyses were run on the online platform (https://bioinfo.mnhn.fr/abi/public/asap/) using Kimura (K80) distance matrices and configured under following parameters: substitution model = p-distances, probability = 0.01, best scores = 10, fixed seed value = -1.

For each statistical method, just provide citation and eventually specific setup. All those methods are known and described by their authors in published articles - citation is good enough. Statements with citations like in rows 256-259, are more suitable for the discussion part of your results.

As a reference of description of consistent methodology, I can recommend to authors an article from Ying-Yuan et al. 2024 (https://doi.org/10.1371/journal.pone.0301776).

Results Molecular analysis of genetic distances I found an issue with sequences named Naphrys sp. - Within the name there are about 2 species: one species is represented by sequence OR174102, and second by sequences OR173350; OR174487; OR174414. It is most likely impossible that in all species of Naphrys intraspecific distances were below 1.61%, except for Naphrys sp. where according to the Table 3 - there is about 10,94% of intraspecific divergence... Four (OR) sequences should be divided into the 2 species: Naphrys sp. 1, sp. 2.
The question is if specimens (sequences)were correctly identified into the Naphrys genus (please check the reference publication if possible).

Furthermore, Maximum likelihood tree (Figure 4), is clearly suggesting that species of N. pulex, N. xerophila, N. echeri, N. tuuca, N. tecoxquin - are representing highly supported clade (monophyletic group?), while specimens N. sp USA 8;9;11 are much distant from. This issue should be precisely described/discussed.

To avoid such issues, I'd like to recommend a more careful selection of available sequences. For the first overview it's good to run and analyse simple delimitation methods and to try to calculate the barcoding gap in alignment. Intraspecific distances within salticid species should be under ca. 2% while intraspecific ones are above 6-10 percent. Having intraspecific distances over 10% means that there are more species in the studied dataset.

Another way of initial COI based species delimitation is RESL method - the one which generate MOTU (Molecular Operational Taxonomic Units) at the BOLD platform, further designed as a BINs (Barcode Index Number - kind of species proxy). In the BOLD platform, there is an option for uploading COI sequences of species/taxon of our interest, and to run simple delimitation analysis. I did such dataset and such calculations - eg. N. pulex is showing high intraspecific divergence and is divided into the 3 BINs - this means = or very high intraspecific divergence (due to distant populations?), or more than one species. It's always worthwhile to start and interpret our analysis first from simple to finally more advanced statistics.

All methodology and former analysis must be clearly described in a precise and detailed way. As I mentioned above, many descriptive rows should be deleted and only citations should be provided while basic results such as COI DNA alignment, should be described in detail - as a base for all analysis.

With the species delimitation results provided in the manuscript, 3 species of Naphrys proposed by authors as a new for science, are clearly new for science.

Taxonomy:

It’s highly valuable that the pictures of habitat/microhabitat are provided, also pictures of alive specimens are important. Jumping spiders preserved in ethanol even after a short time are losing natural body coloration – microphotographs of alive specimens should be present in taxonomical articles (if possible).

N. acerba - why there are tables and drawings of this species???

I can't find any information (in Abstract, Introduction), about the issue of the Naphrys acerba species - in the section of taxonomy I found that this species is redescribed? New diagnosis, new description, pictures, drawings - it must be emphasised in the aims of an article. Redescription is very important and useful - the question is if authors are sure that they redescribed the proper species? It's difficult without studying type collection for a species, sometimes it is possible, but must be explained.

Genus distribution: Naphrys bufoides was recorded in Chile due to data provided from the most useful catalogue of jumping spiders (https://www.jumping-spiders.com/index.php?/nav_distribution/verb_set.html) is that true or this is just a mistake in the catalogue data?

All Diagnosis provided for newly described or redescribed species are too long. In the diagnosis, species should be compared with another - the most morphologically similar species – to emphasize all differences. Diagnosis is not the place for comparing all species. Description from diagnosis shouldn’t be repeated in the description.

Descriptions: first provide the description and all abbreviations with measurements at the end. In methods, abbreviations were provided, there’s no need to explain again that for example “Anterior eyes row AER 1.47 times wider than PER”.

Diagnosis and descriptions must be precise and consistent.

Authors provided first such a detailed drawing, LEM and SEM pictures for the genus Naphrys giving new insight into the ultra-morphological structures of the Naphrys male/female copulatory organs. This opportunity allows authors to consider some redefinition of copulatory organs structures in the genus.
Looking at the structures and shape of some structures in male and female copulatory organs, I may suggest to authors that structures on epigyne in Naphrys, called Copulatory duct (CD) , are not copulatory ducts in Naphrys. Copulatory duct should have some channel-like structure, connected directly with spermatecae.
In the genus Naphrys, structures called CD are looking like some shields, supporting the embolus to connect with CO/CD – which is short and directly connected with
spermatecae.

For example, on the picture of epigyne on Figure 10C, authors marked some dark brown structures as CO – this structure might be considered as a short copulatory duct.

Furthermore, shields around the epigynal window may most likely correspond with shape of embolus – and may play supportive function to the embolus, during the copulation (Lock and key mechanisms - genitalic evolution/coevolution).

On the drawing of epigyne, authors already have distinguished some different (intriguing) structures (Figure 10E-F), transparent thin, sclerotized shield (marked as CD), and strongly sclerotized, getting wider to the spermatecae direction structure called CO. In my opinion CD on Fig. 10 is just a sclerotized shield supporting the embolus, CO is a copulatory duct with copulatory opening (CO in general should be placed at the entrance of the copulatory duct – this supports my hypothesis). Combination may correspond to all species of the genus Naphrys. In addition, spermatecae usually have accessory glands (many times omitted in the descriptions) – if I’m correct, on the pictures and drawings of epigyne (Figure 15C&E) such structures like accessory glands are visible near the Fertilization ducts.

It's possible that I’m wrong with the matter of redefinition of the copulatory organs in Naphrys genus, but always starting from the species descriptions, it is important to consider all structures under the evolutionary point of view. I’ll leave this suggestion to the authors.

Validity of the findings

no comment

Additional comments

As I mentioned at the beginning, the manuscript is important and interesting for the knowledge on jumping spiders, but all issues mentioned above must be clarified before an article is accepted and published. I’d like to get feedback from authors and to check an article again. The results (except some details I explained above) are solid and of a high value.

This is a good and important manuscript fitting well in the scope of the journal. To be sure that my suggestions about making some sections shorter, precise, and clearer, I recommend major revision.

I'll be very happy to see the corrected and finally published manuscript, and with pleasure may serve in further assistance if needed.

·

Basic reporting

This is an impressive paper in many regards. The morphological work is detailed, well-presented, informative, and important. The diagnoses are well done — this is often a weak point of taxonomic papers, but not in this one. The key is a good addition. The species are clearly new by their morphology. The paper reveals an unappreciated radiation of the genus much further south than known before. The molecular data and analysis are a useful addition. The writing is articulate and well-organized.

I have very few concerns about the substance of the paper. My concerns relate to presentation. The most important substantive request is to show better images of the leg colours of males (see below, "Leg colours").

Experimental design

The morphological and molecular data are well presented and interpreted.

Validity of the findings

The findings of three new species are well supported. In addition, many other findings about these species -- in the form of information about their morphology, natural history and distribution -- are given and well documented.

Additional comments

In my view, the primary value of this paper, and the bulk of the work behind it, is in the exemplary morphological taxonomy. It is definitely worth publishing for that alone.

However, the emphasis in the abstract, introduction, and discussion is on a different topic, the use of molecular data. The paper gives a thorough argument for the value of using genetic methods for species delimitation in addition to morphological approaches. The Discussion devotes 3 full pages to considering the use of molecular species delimitation. However, the argument should be seen primarily for its theoretical value, as the paper itself does not strongly exemplify the approach, because only a single gene (COI) is used. A single linkage group, particularly one like COI known to introgress, is a weak indicator of species limits. The standard for molecular species delimitation studies has shifted to hundreds of genes, and for good reason — one or two genes are often discordant with others, as expected by coalescence theory.

The relative strength of the evidence from morphology vs. molecules in Naphrys can be expressed like this: if there had been a contradiction between the results of the morphology and that of COI, I would have had more confidence in the morphology. For example, in the face of the clear morphological differences, if the COI had not shown clear distinctions, it would have been most reasonable to suppose distinct species but a high level of ILS. (If 1000 genes had shown no pattern, it would be a different story.) If the morphology had shown no differences, but the COI clustered clearly as found, it would have been most reasonable to assume population structure rather than species distinctions (e.g., see Sukumaran & Knowles 2017). Either way, the morphological evidence is stronger.

Please note that these comments are not a criticism of the substance of the data and the paper's interpretation about these species. Rather, my comments concern the presentation, which spends too many words on the value of the molecular approach. That is not to say that the molecular data presented here have no value, just that the very lengthy rhetoric for the approach is disproportionate to its value in this case. That is especially true given that the clear differences in morphology leave little doubt of the distinction of the species anyway.

Comments on the Abstract:
The abstract, like the paper, emphasizes the use of DNA for species delimitation. This I see as an inappropriate emphasis, because, because, as I argue above, the primary value of the paper is in the morphology.

However, even though it's emphasized, what is said about the molecular analysis isn't complete in the abstract. There are two components to the molecular study: the data gathered, and the analyses done. While the Abstract goes into great details about analysis done, it says nothing about the data. It should. Abstracts are expected to describe the nature of the data — COI? 4 loci Sanger-sequenced? UCE data? Low coverage genomes? The answer here is COI, and the abstract should say so.

Comments on the paper otherwise:
Hedin and Milne (2023) might best be mentioned -- this is an exemplary study of integrative taxonomy in spiders: https://zookeys.pensoft.net/article/96724/element/5/31/

Palp illustrations: The photos/drawings of palps are not all oriented correctly. (This is not a request for a change, but it is worth emphasizing for future consideration.) The N. acerba and N. tuuca palps ventral view are tilted to give a slightly retrolateral view. The N. echeri palp ventral view is tilted to give a slightly distal view. The N. tecoxquin is good in orientation. It would help show differences of the palps are shown in the exact same orientation.

Leg colours: The species are very distinctive in their markings in males, which facilitates field identification. The striped nature of N. tuuca is well shown in the living photo, for example. However, one difference appears strong but is not well-illustrated: the colours of the legs of males. N. tuuca has the first and second legs pale. N. tecozquin has the first and second legs much darker than the third and fourth in Figure 14A, but only a small sliver of the leg is visible. With N. echeri it is difficult to say what are the leg colours. Given how helpful these differences are for identification, would it be possible to show the complete legs on one side so that this contrast is more visible to readers? Also, these clear differences in legs are not mentioned in the diagnoses.

There is no longer a recognized status of "allotype". Such a specimen is just a paratype of the opposite sex from the holotype. The other paratypes of the opposite sex from the holotype have the exact same status as the "allotype".

Note that figure numbers in the text are incorrect. E.g., Fig 7D mentioned under the male of N. echeri in the key is actually of the female of N. acerba. I did not check to see what other errors there might be.

I have annotated the PDF with corrections and comments.

---

## Round 0.2 · Minor Revisions

· Academic Editor

Minor Revisions

The authors need to include the final very minor reviewer comments.

·

Basic reporting

The manuscript will be ready for publication as soon as the authors make minor adjustments to pages 14, 17, 19, 84.

Experimental design

Ok.

Validity of the findings

Ok.

·

Basic reporting

Dear Editor,
Dear Authors,

I'm satisfied that the Authors have responded to most of my comments.

Last few concerns below:

PCR products were purified at Psomagen, followed by Sanger sequencing in both forward and reverse directions. To clarify any doubts, this is specified in the text (L180). The sequences obtained were subjected to manual editing and then aligned (see Sequence editing and alignment section). Non-informative regions were excluded from the final sequence, which was then submitted to GenBank and found to be 589 base pairs in length.

I'm not sure what are the "non-informative" regions? The COI gene fragment after amplification and sequencing with the standard Folmer primers, should have about 648 base pairs. Usually sequencing in both forward and reverse directions allow us to get the full 648bp length sequence, where all nucleotides are informative. In general, we're removing only the sequence of primers, eventually some longer fragments at the beginning and at the end of the sequence, when there is some noise rather than proper nucleotides.

All long, detailed descriptions of DNA analysis (Methods) are not necessary in such cases, but I understand that the authors: "chosen to retain a detailed description for those unfamiliar with the methods". It might be useful for some other authors.

I'm happy that the Authors provided molecular data here - even COI - standard barcode fragment is important while still most of the articles on spider taxonomy really on the morphology only - such integrative taxonomy is needed nowadays.

In my opinion providing a full description of the measured character both with shortcut “Anterior eyes row AER 1.47 times wider than PER” is a mistake.

I provided some doubts into the description of female/male copulatory organs morphology, and still have some impression that those have different nature than described, but the Authors are specialists in the jumping spider genus Naphrys, and as authors of new species descriptions, are responsible for their accuracy.

Finally, the manuscript is looking much better. I also like the discussion - many useful references can be found there.

I believe that the manuscript entitled: “Three new species of the spider genus Naphrys Edwards (Araneae, Salticidae) under morphology and molecular data with notes in the distribution of Naphrys acerba (Peckham & Peckham) from Mexico” is ready to be published in PeerJ.

Kind regards,

Łukasz Trębicki

Experimental design

no comment

Validity of the findings

no comment

---

## Round 0.3 · accepted · Accept

· Academic Editor

Accept

The authors should check the references carefully while in production.

·

Basic reporting

The manuscript can be accepted for publication as it brings new insights to the field of arachnology, particularly from the perspective of a neglected area in the sciences, such as taxonomy. Reviewing works on this topic, using integrated data and collaboration among the authors, advances scientific knowledge.

Experimental design

Ok

Validity of the findings

Ok

Additional comments

Ok